# Sustainable-use marine protected areas to improve human nutrition

**Daniel F. Viana** [1,2] ✉, **David Gill**[3], **Alex Zvoleff** [4], **Nils C. Krueck** [5], **Jessica Zamborain-Mason** [1,6], **Christopher M. Free** [7,8], **Alon Shepon** [9], **Dana Grieco** [3], **Josef Schmidhuber**[10], **Michael B. Mascia**[4,11] & **Christopher D. Golden** [1,6,12]

Coral reef fisheries are a vital source of nutrients for thousands of nutritionally vulnerable coastal communities around the world. Marine protected areas are regions of the ocean designed to preserve or rehabilitate marine ecosystems and thereby increase reef fish biomass. Here, we evaluate the potential effects of expanding a subset of marine protected areas that allow some level of fishing within their borders (sustainable-use MPAs) to improve the nutrition of coastal communities. We estimate that, depending on site characteristics, expanding sustainable-use MPAs could increase catch by up to 20%, which could help prevent 0.3-2.85 million cases of inadequate micronutrient intake in coral reef nations. Our study highlights the potential add-on nutritional benefits of expanding sustainable-use MPAs in coral reef regions and pinpoints locations with the greatest potential to reduce inadequate micronutrient intake level. These findings provide critical knowledge given international momentum to cover 30% of the ocean with MPAs by 2030 and eradicate malnutrition in all its forms.

Over 2 billion people are unable to access safe, nutritious, and sufficient supplies of food, which threatens human health globally[1]. Millions of coastal residents in tropical developing countries rely on fisheries resources as a vital source of minerals, vitamins, and fatty acids[2,3], and are particularly vulnerable to nutritional deficiencies[4–6]. For many of these coastal populations, coral reef systems are a critical source of nutrients and livelihoods[7]. Yet, coral reefs around the world are being severely degraded by pollution, overfishing, and climate change, imperiling marine biodiversity and human health[8]. Policies governing coral reefs that attempt to address these threats not only shape the future of these ecosystems but also the health of people who depend upon them.

Marine protected areas (MPAs) are increasingly used to protect coral reef ecosystems and recover associated fisheries from depletion globally[9]. MPAs are areas of the ocean with specific rules and restrictions primarily designed to protect marine ecosystems from anthropogenic threats[10,11]. MPAs can be broadly categorized into no-take areas where all fishing is strictly prohibited and partially protected or sustainable-use areas where some forms of fishing are permitted. Sustainable-use MPAs use various fisheries management tools to support fisheries sustainability and may be zoned for multiple uses (e.g.,

[1]Department of Nutrition, Harvard T.H. Chan School of Public Health, Boston, MA 02115, USA. [2]Ocean Conservation, World Wildlife Fund, Washington, DC 20037, USA. [3]Duke University Marine Laboratory, Nicholas School of the Environment, Duke University, Beaufort, NC 28516, USA. [4]Moore Center for Science, Conservation International, Arlington, VA, USA. [5]Institute for Marine and Antarctic Studies (IMAS), University of Tasmania, Hobart, Tasmania 7001, Australia. [6]Department of Environmental Health, Harvard T.H. Chan School of Public Health, Boston, MA 02115, USA. [7]Bren School of Environmental Science and Management, University of California, Santa Barbara, Santa Barbara, CA, USA. [8]Marine Sciences Institute, University of California, Santa Barbara, CA, USA. [9]Department of Environmental Studies, The Porter School of the Environment and Earth Sciences, Tel Aviv University, Tel Aviv, Israel. [10]Pure Harvest Smart Farms, Dubai, United Arab Emirates. [11]Sanford School of Public Policy, Duke University, Durham, NC, USA. [12]Department of Global Health and Population, Harvard T.H. Chan School of Public Health, Boston, MA 02115, USA. ✉e-mail: dviana@hsph.harvard.edu

tourism, aquaculture, conservation). Both MPA types generally have significant positive effects on the biomass of targeted fish within their boundaries compared to neighboring non-MPA areas[12]. Currently, there are about seventeen thousand MPAs worldwide, covering about 8.2% of the ocean[13]. Despite failing to reach the globally agreed target of 10% MPA coverage by 2020[14], there is a new global commitment to cover 30% of our oceans with MPAs and other effective area-based conservation measures (OECMs) by 2030[14–16].

The integration of sustainable-use MPAs into the food and nutrition agendas of coral reef countries is often overlooked[17]. While good fisheries management holds the potential to enhance human nutrition[6], its success is contingent upon the availability of high-quality data and robust management capacities. Sustainable-use MPAs emerge as a realistic and cost-effective intervention in various settings, offering a balance between conservation goals and sustainable resource utilization[18]. Moreover, these conservation measures play a crucial role in safeguarding the rights of local communities, protecting them from exploitation by non-local entities[19]. This not only underscores the significance of MPAs as a pivotal tool in sustainable resource management but also highlights their potential multifaceted impacts on both ecological conservation and human nutrition.

Here, we quantified the potential add-on human nutritional benefits that can arise from sustainable-use MPA implementation through increases in local reef fish catch and consumption. We started by compiling information on coral reef fish populations and social and environmental conditions from 2421 reef sites (1117 no-take MPA, 804 sustainable-use MPA, 500 non-MPA) in 53 countries to estimate the effect of these areas on standing reef fish biomass. We then used a Bayesian hierarchical model to estimate the expected standing reef fish biomass under non-MPA and sustainable-use MPAs conditions, accounting for other social (e.g., human population, market distance, fisheries governance, human development index) and environmental (e.g., productivity, depth, temperature, wave exposure) variables (see Supplementary Material for details). We then used the model to estimate: (1) the expected biomass and catch for all existing sustainable-use MPAs; (2) the potential biomass and catch associated with an expansion of sustainable-use MPAs to all non-MPA reefs; (3) the change in nutrient supply due to changes in coral reef fish catch; and (4) the potential changes in nutritional inadequacies associated with sustainable-use MPA expansion (see Supplementary Fig. M1 for details). We focused our analyses on zinc, iron, calcium, omega-3 long-chain polyunsaturated fatty acids docosahexaenoic acid (DHA) and eicosapentaenoic acid (EPA) (hereafter referred as DHA + EPA), and

vitamins A and $B_{12}$, which are abundant in aquatic species and critical for human health[6].

## Results

### Conservation benefits of sustainable-use MPAs

We estimated the potential net conservation benefits of sustainable-use MPA establishment by examining the effect of sustainable-use MPAs on reef fish biomass (Fig. 1). Biomass estimates are based on underwater surveys from around the world, with most observations from Australia, the Caribbean and Southwest Pacific (Fig. 1A, B). Globally, we found that sustainable-use MPAs have on average 15% more biomass than non-MPA sites, although differences were highly dependent on the effectiveness of fisheries management in surrounding waters (Fig. 1C). Estimated differences in biomass were calculated based on the effect size of sustainable-use MPAs on reef fish biomass. Locations with high fisheries management effectiveness (FME; Fig. 1C) had a lower difference in biomass between non-MPA and sustainable-use MPA biomass relative to locations with low management effectiveness, likely because non-MPA sites in high FME locations are already well-managed and therefore less depleted.

### Nutritional benefits from existing sustainable-use MPAs

We estimated the current biomass and associated catch increases from existing sustainable-use MPAs by comparing their estimated catch to the expected catch at those sites if they were not sustainable-use MPAs. Using a Schaefer model to simulate population dynamics[20], this analysis was completed in three steps (see Supplementary Material for details). First, we used our hierarchical Bayesian model, which accounted for varying social and environmental contexts, to estimate the biomass of the site if it were not a sustainable-use MPA. Second, we derived the status of each site, defined as the ratio of current biomass (observed in the sustainable-use scenario and predicted in the non-MPA scenario) to the carrying capacity (see Supplementary Material for details). Finally, we derived the long-term catch under equilibrium assuming a community-wide level of productivity (i.e., a multispecies intrinsic growth rate of 0.23)[21]. We estimated that existing sustainable-use MPAs provide an average of 12% more catch than they would if they were not MPAs.

Expanding seafood production and consumption is an impactful pathway for addressing nutrient inadequacy[6]. Many existing sustainable-use MPAs are in areas with large coastal populations at high risk of inadequate nutrient intake[6], such as Indonesia, the Philippines, and Haiti, where expected improvements in catch are the

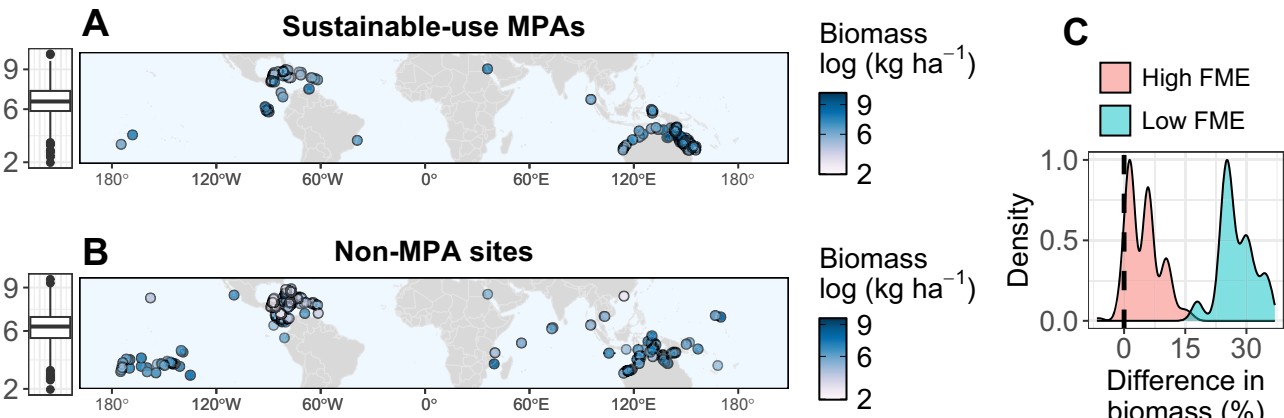

**Fig. 1 | Observed biomass and estimated effect size of sustainable-use marine protected areas.** Maps show **A** observed biomass in sustainable-use MPA sites (n = 804), and **B** observed biomass in non-MPA sites (n = 1117). **C** highlights distribution of percent differences in biomass of sustainable-use MPAs compared to non-MPA sites for locations with high (>0.5) and low (≤0.5) estimated fisheries management effectiveness (FME). Differences in biomass are the estimated effect size of sustainable-use MPAs relative to non-MPA sites.

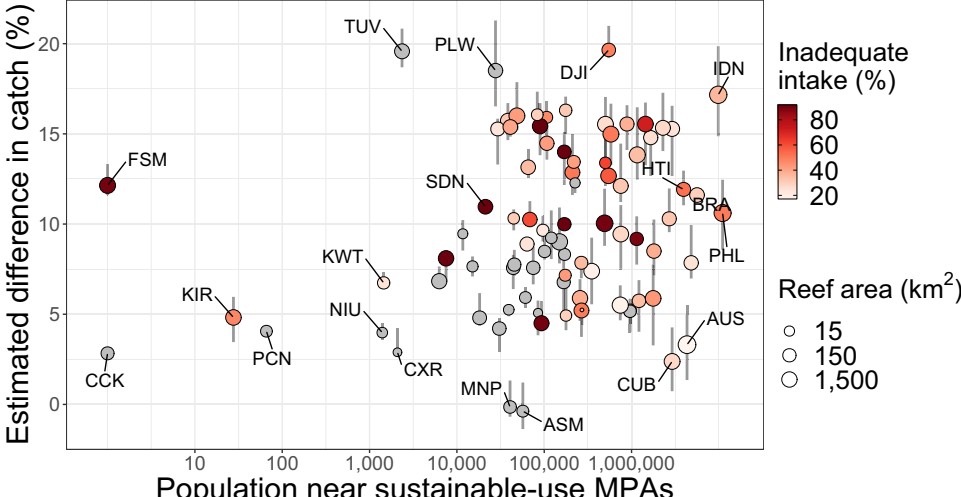

**Fig. 2 | Estimated catch and nutritional benefits from existing sustainable-use MPAs.** Estimated catch effects in existing sustainable-use MPAs relative to expected catch under non-MPA conditions, plotted on a log-scale for country populations within 10 km of sustainable-use MPAs. Error lines represent the variability across reef polygons within each country. Inadequate nutrient intake is the average prevalence across key nutrients found in aquatic species in 2017 (iron, EPA + DHA, calcium, zinc, and vitamins A and B$_{12}$). Points in gray represent countries with no data on the prevalence of nutrient intake. Three-letter abbreviations represent selected alpha-3 ISO country codes.

highest (Fig. 2). Inadequate intake values range from 0% to 100% and should be considered as the human populations' risk of nutritional inadequacies, with higher values representing larger populations at risk of inadequate micronutrient intake[22].

**Expanding sustainable-use MPAs to improve human nutrition**

Overlaying MPA boundary data from MPAtlas[13] with spatial data on reef area[23], we found that 37% of all coral reefs in the world are within sustainable-use MPAs, 11% are within no-take MPAs, and 51% are non-MPA reefs (Supplementary Fig. S1). We predicted the potential human nutritional benefits of expanding sustainable-use MPAs to all non-MPA reefs (~70,000 km², Fig. 3A), by (i) calculating the potential increase in biomass and thus catch from sustainable-use MPA expansion, (ii) estimating the total number of people that could be nutritionally supported by existing and future sustainable-use MPAs, and (iii) calculating the potential change in the risk of prevalence of inadequate nutrient intake with sustainable-use MPA expansion (see Supplementary Fig. M1 in supplementary methods for details). We defined nutritional support from sustainable-use MPAs as the provision of at least 5% of aquatic animal source food intake from coral reef catch (see Supplementary Fig. S8 for sensitivity analysis). We then summed the population near reefs (10 km) above a range of threshold (5–30%) to estimate the total number of people supported by sustainable-use MPAs. We found that sustainable-use MPAs could support the nutritional needs of 2.8–30 million people by substantially contributing to overall aquatic animal source food intake.

Next, we calculated how changes in catch due to sustainable-use MPA expansion could affect nutrient supply (Fig. 3B). We calculated changes in catch by comparing predicted biomass and catch from non-MPA reefs if they were in sustainable-use MPAs versus non-MPA and estimated overall nutrient supply from the potential catch. Differences in nutrient supply attributed to predicted changes in catch were then added to the overall diets of populations living within a 10 km buffer around reefs (see Supplement Material for sensitivity analysis) to predict the potential changes in inadequate nutrient intake risk from sustainable-use MPA expansion. We found that expanding sustainable-use MPAs to non-MPA locations could increase catch in many nutritionally vulnerable countries (i.e., populations with inadequate intakes higher than 25% across all selected nutrients), with potential positive impacts on human nutrition and health. On average, catch could

increase by 12% when considering all countries or 15% (from 2–20%) if we only consider nutritionally vulnerable countries.

Overall, four major factors drive the extent of potential nutritional impacts of sustainable-use MPA expansion in our study: (i) non-MPA reef area (Fig. 3A), (ii) population size near non-MPA coral reefs (Fig. 3C), (iii) the prevalence of inadequate intake within coastal communities (Fig. 3E), and (iv) the efficacy of local fisheries management. First, the larger the area of a non-MPA reef (Fig. 3A), the larger the potential for sustainable-use MPA expansion to provide nutritional benefits. Second, the size of the local population around reefs (Fig. 3C) determines the per capita consumption estimate for reef-caught seafood (Fig. 3D). While larger local populations will lead to lower per capita impacts, with potentially higher numbers of people impacted, smaller populations can have higher per capita impacts. Third, increasing catches in coastal communities with high levels of inadequate intake (Fig. 4E) has the greatest potential to decrease nutritional risks for nutrients that are abundant in reef fish. On the other hand, increasing catch in communities that already have adequate nutrient intake will have minimal impact on nutritional health. Lastly, the data supported the addition of an interaction between sustainable-use MPA and the national efficacy of fisheries management in the model (Supplementary Fig. S2), suggesting that, as expected, changes in catch following sustainable-use MPA establishment will depend on the strength of local fisheries management in surrounding areas[24]. This reflects the fact that locations with high fisheries management efficacy, which may have high biomass outside of sustainable-use MPAs, have little or no potential for sustainable-use MPAs to provide net biomass increases. Because of uncertainty around these four factors, the absolute number of people impacted by sustainable-use MPA expansion is also uncertain (See Supplementary Fig. S8 for sensitivity analysis). Yet, our results provide strong support for the hypothesis that MPAs can benefit human nutrition, and the general geographical patterns appear robust. Countries such as Madagascar, Mozambique, Kiribati, Yemen, and the Solomon Islands have the highest potential reductions of inadequate nutrient intake. Other countries, including Seychelles and Sudan, have similarly high potential changes in per capita seafood availability but cannot be modeled in terms of inadequate intake because of a lack of catch or baseline nutrient supply data.

Globally, the expansion of sustainable-use MPAs could lead to reductions in inadequate intake across all assessed nutrients for

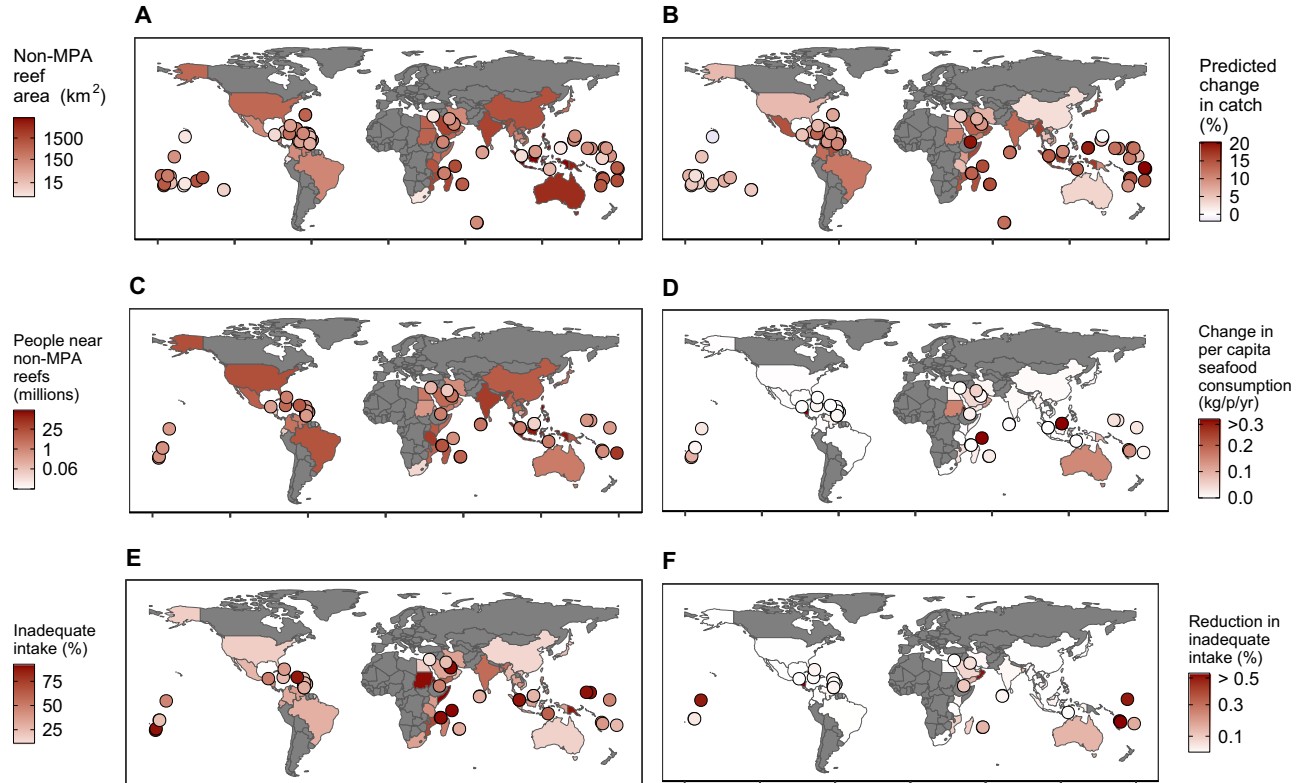

**Fig. 3 | Potential nutritional impacts of expanding sustainable-use MPAs into unprotected reefs.** Maps show (**A**) non-MPA reef area within each country based on the overlap between MPAs (sustainable-use and no-take) and reef areas; **B** predicted change in catch in a hypothetical sustainable-use MPA expansion into non-MPA reefs; **C** total number of people within 10 km of non-MPA reefs; **D** predicted change in per capita consumption of seafood for the population around non-MPA reefs; **E** prevalence of present inadequate intake across countries; and **F** predicted average change in inadequate intake for coral reef coastal populations across all assessed nutrients (iron, EPA + DHA, calcium, zinc, and vitamins A and B$_{12}$). Countries smaller than 25,000 km$^2$ are illustrated as points.

0.3–2.85 million individuals (reduction of 0.2–1.5 million for vitamin B$_{12}$, 0.1–0.7 million for calcium, 0.05–0.5 million for omega-3 long-chain polyunsaturated fatty acids (specifically DHA + EPA), 0.07–0.4 million for iron, 0.06–0.2 million for vitamin A, and 0.02–0.15 million for zinc inadequate intakes) (Fig. 4A). Such reduction in inadequate intake represents about 1–7% of the combined total nutritional benefit that reforms in all marine wild capture fisheries could be expected to support[6]. Regions such as South and Southeast Asia, Pacific, Latin America and Caribbean (see Supplementary Table 2 for a complete list of countries) have the highest potential for nutritional improvement following implementation of effective sustainable-use MPAs (Fig. 4A). For other regions such as Sub-Saharan Africa, sustainable-use MPAs can also have important effects at a local level, but on aggregate at national scales, total impact is relatively low due to reduced total reef area and lower numbers of people living around reefs.

### Nutritional targeting of vulnerable populations

Because each country has different specific nutrient inadequacies, sustainable MPAs can be strategically placed in areas with the highest potential to reduce specific nutritional inadequacies (Fig. 4B). For example, creating targeted sustainable-use MPAs in Yemen and Kenya has the potential to reduce inadequate intake risks of omega-3 long-chain polyunsaturated fatty acids (DHA + EPA). Increased intake of DHA + EPA can promote brain and eye health and is associated with reduced risk of heart disease[25]. India and Bangladesh could particularly benefit from increased supply of vitamin B$_{12}$, where inadequacy is more than twice the global average[6]. Vitamin B$_{12}$ deficiency is associated with increased risk of heart disease and cognitive decline[26]. Mozambique and Cambodia could benefit from increased supply of iron, which is particularly important for healthy brain development

and growth in children[27], and can prevent maternal mortality[25]. Sustainable-use MPAs in Nicaragua and Madagascar have the potential to reduce inadequate intake of zinc, which supports immunity and is particularly important for children and pregnant women[28]. Kuwait and Indonesia could benefit mostly from increased supply of calcium, which supports bone health and blood pressure[29]. Lastly, Oman and Kiribati could benefit from increased vitamin A supply, which supports eye health and cell growth[30].

## Discussion

Our analysis suggests that effective sustainable-use MPAs have the potential to increase biomass and support human nutritional security through increased sustainable catches from coral reef ecosystems. These benefits are on top of the documented MPA impacts on livelihoods[11,31] and marine biodiversity[12,32] (in essence, co-benefits). We predict that expansion of sustainable-use MPA to non-MPA coral reefs can have important nutritional benefits by both sustaining the currently adequate nutrition of coastal populations and decreasing the prevalence of inadequate intake of vital nutrients of vulnerable people. For coastal communities that rely on coral reef resources for key nutrient intake, increasing sustainable supply of nutritious food can have positive impacts on their health and well-being.

Many factors determine the magnitude of impacts on a local scale. These factors are related to the availability, access, and utilization of reef fish caught from sustainable-use MPAs. For example, our analysis suggests that availability will be influenced by local factors such as the productivity of the local reef system, the number of seafood consumers (which impacts per capita consumption), the prevalence of inadequate intake, and the efficacy of local fisheries management. However, for additional catch through sustainable-use MPA expansion

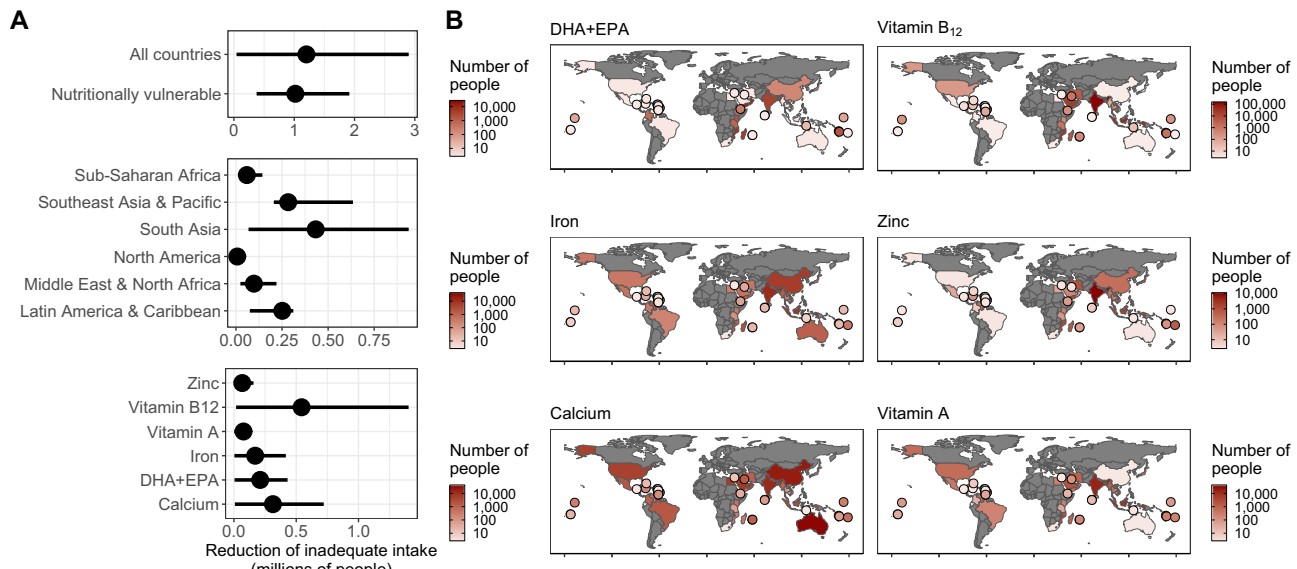

**Fig. 4 | Potential nutritional benefits from sustainable-use MPA expansion.**
Total reduction (in number of people) in the risk of inadequate intake of key nutrients from coral reef sources for coastal human populations attributed to the expansion of sustainable-use MPAs relative to non-MPA conditions. **A** Total reduction in inadequate intake risk by country type, across regions and across nutrients. Nutritionally vulnerable include countries with inadequate intakes higher than 25% across all selected nutrients. The range of values represents uncertainty around estimates based on Monte Carlo simulation and points represent median values from simulations. **B** Country-level impacts for each assessed nutrient. Countries smaller than 25,000 km² are illustrated as points.

to have an impact on nutritional health, it is important that those in need of nutrients have access to increased catch from sustainable-use MPAs. In cases where communities already have high consumption of fish[33], sustainable-use MPAs can be key to maintain availability and access to nutrients from coral reefs. With seafood being one of the most widely traded food commodities in the world[34,35], it is important to ensure that sustainable-use MPA catch is affordable and accessible for domestic consumption[36]. In addition, cultural and dietary preferences and fish utilization practices can also influence nutritional impacts. Factors such as sanitary conditions of the fish caught and sold, and seafood loss and waste can also fundamentally influence shelf life, nutritional quality, and availability of reef fish[37].

Our model predictions are based on the current performance of sustainable-use MPAs. To restrict catch, these areas use an array of fisheries management tools, including gear restrictions, access rights, size limits, temporal closures, bag limits, and more[9]. Some sustainable-use MPAs may be zoned for multiple uses, potentially containing small no-take areas, which may benefit fished areas through spillover[38]. Because our results depend on the actual performance of sustainable-use MPAs, if management of these areas improves (through more investment, management capacity, planning, community participation, etc.), potential biomass increases and consequential nutritional gains could be even greater. Understanding how close sustainable-use MPAs are to their maximum sustainable yield can provide a better estimate of their true potential to provide nutritional benefits. In some cases, reef stocks could be managed to maximize specific nutritional yields, tailored to the needs of local human populations[39]. However, for fish stocks experiencing high fishing pressure and depletion below the biomass supporting maximum sustainable yields, recovery will likely require a short-term decrease in catch to obtain long-term gains[40,41]. Therefore, it is important to consider the impacts of potential short-term economic and nutritional costs to achieve predicted long-term nutritional benefits. Given that fish recovery may require restricting catch for an extended period, policy makers should seek to mitigate impacts of regulations on nutritional security in highly dependent populations.

We consider sustainable-use MPAs as all protected areas where some form of fishing is permitted[10]. In addition to state-managed areas,

this definition encompasses many types of area-based management strategies co-managed with or solely managed by local communities. Different from top-down fisheries management strategies, these and some other effective area-based conservation measures (OECMs) can empower local stakeholders and incentivize sustainable resource management, while considering local characteristics of the fishery and cultural traditions of coastal communities[42]. Several studies have shown that when communities are empowered and have secure rights to a fishery, there is greater incentive for successful fisheries management yielding nutritional benefits[43–45]. However, more research is needed to evaluate how different types of sustainable-use MPAs and restrictions affect biomass, catch, and human nutrition. Various other management measures (e.g., individual quotas) that are implemented outside of sustainable-use MPAs could also be effective at improving nutrient supply. Our analysis suggests that fisheries management effectiveness at a national level has a strong effect on reef fish biomass and can be important to providing broad nutritional benefits to coastal populations.

Nutritional benefits of sustainable-use MPAs should be viewed as an add-on benefit that supplements the marine conservation goals of MPAs. Compared to other measures to address malnutrition, sustainable-use MPAs can be costly and take a long time to provide desired benefits. Other measures such as nutritional supplements and agriculture development can be faster and more cost-effective to address malnutrition[46]. However, sustainable-use MPAs represent a synergistic policy that can meet multiple objectives: conservation, food security, ecosystem resilience, property rights allocation, conflict alleviation, etc., while other targeted policies are usually for single purposes. In addition, sustainable-use MPAs can act as a safety net during food supply shocks, providing a reliable nutrient source easily accessible by local communities. Moreover, sustainable-use MPAs can be important to mitigate future nutritional loss due to anthropogenic actions such as overfishing and climate change[8].

Climate change impacts on reef ecosystems create an uncertain future for millions of people who depend on reef fisheries for nutrition and livelihood[8]. Without actions (such as sustainable-use MPAs) to ensure sustainability of catch into the future, there could be significant losses of nutritional benefits in the coming years, with important

implications for public health. Therefore, it is not only the prevention of inadequate intake and the supporting benefits of sustainable-use MPAs that can be important, but also the buffering against the risk of future loss in climate-vulnerable systems. Without conservation action in the present, the risk of future negative nutritional impacts is inevitable in the long term[8].

Expansion of sustainable-use MPAs depends on strong local, national, and global commitments and investments. Today, only a small fraction of resources from governments, regional development banks, and multilateral funding agencies are directed to strengthening governance of small-scale fisheries[47]. The lack of adequate capacity to manage sustainable-use MPAs has led to the creation of many paper parks, which are designated areas that are not effectively implemented[10,12,48], thus having limited potential to provide environmental, economic, and nutritional benefits. Our results suggest that populations who depend on reef systems for nutrition would benefit from directed resources to sustainable marine resource management and to ensuring that harvested seafood is safe to eat and accessible to those in need.

## Methods
### Reef fish underwater surveys
We compiled information of coral reef transects from the Reef Life Survey (RLS)[49] and Atlantic and Gulf Rapid Reef Assessment (AGRRA) databases[50]. Both databases are based on underwater fish counts by size class within a belt transect conducted from 1997 to 2020. We then calculated individual biomass by using length-weight relationships published for all species on Fishbase[51] and then multiplying individual biomass by the total number of fish within each size class. The final compiled database contained 16,365 surveys from 2421 tropical coral reef sites (i.e., within 23.5 latitude degrees) distributed across 53 countries. Where data from multiple years were available for a single site, we included only the most recent year. To estimate the "fishable biomass" we retained only fish larger than 10 cm[52]. Because underwater fish counts do not accurately capture biomass of large schools of pelagic fish (e.g., Scombrids, Sphyraenids) or large transient fish (including shark and ray species), we removed them from the analysis[52,53]. Depending on the site, pelagic and large transient fish can be important for biomass, catch, and nutrition. Based on the fish captured in the underwater surveys, pelagic and transient fish represented a median of 15% of the total biomass across all sample sites. In addition, because of data constraints, we only considered reef fish catch as a nutrient source, however, invertebrate species and aquatic plants can also be an important source of nutrients in many low-income countries. Finally, underwater visual surveys can be affected by the behavior of different fish species and size of larger fish tend to be underestimated[53]. Therefore, results presented here are likely an underestimate of the total nutritional benefits that sustainable-use MPAs can provide to local populations.

Within each database, all survey sites are divided into three basic categories: non-MPA areas, sustainable-use MPAs, and no-take MPAs as defined by the MPAtlas database[13]. Non-MPA areas are all sites located outside of marine protected areas. Although these sites are subject to regional or national-level policies (whether enforced or not), they are not generally managed through additional area-based regulations. Sustainable-use MPA sites are all sites within an area-based management system that allows fishing within its borders, including areas such as multiple-use MPAs, Locally Managed Marine Areas (LMMAs), or Environmental Protection Areas (EPAs). No-take MPAs, in contrast, describe areas where no forms of fishing are allowed (also known as fully protected, marine refugia, etc.). In total, we had 1117 non-MPA sites, 804 sustainable-use MPA sites, and 500 no-take MPA sites.

### Spatial analysis
For each coral reef polygon, we calculated the total reef area that is within a sustainable-use MPA, no-take MPA, and outside MPAs. To do

this, we intersected all coral reef polygons[23] with MPA polygons from the MPAtlas[13]. Within the MPAtlas database, each MPA was divided into sustainable-use or no-take MPAs, allowing the calculation of the percentage of reefs falling within each category.

Next, we calculated the population around existing MPAs and non-MPA reefs by intersecting all reef and MPA polygons with the raster of the Gridded Population of the World in 2019[54]. We calculated the population within 5, 10, 20, 25, and 30 kilometer buffers around reefs and MPAs. To avoid double-counting coastal populations, all overlapping MPA polygon buffers were aggregated. We used the **sf** package[55] in R statistical software[56] to perform all spatial analysis.

### Predicting fish biomass
We used a two-level linear Bayesian model with a normal error structure to predict log reef fish biomass (above 10 cm) in every coral reef around the world based on reef fish biomass observations. For all coral reef polygons, we predicted the biomass of reef fish (excluding pelagic or transient species) per unit area (kg/ha) under two alternative conditions: non-MPA and sustainable-use MPAs, while accounting for each site's own environmental and social covariates (Supplementary Fig. S2). Site covariates considered in this analysis included chlorophyll concentration, sea surface temperature mean, sea surface temperature range, nitrate concentration, wave exposure, reef area, shore distance, human population, market distance, human development index, government effectiveness, and fisheries management effectiveness (see Table S1 and Supplementary Material for detailed descriptions)[52,57]. To account for variability in MPA effectiveness across countries (due to differences in management, staff capacity, state of the reef prior to MPA establishment, etc.), we also considered an interaction term between the presence of sustainable-use MPAs and fisheries management effectiveness (FME) across nations[24]. FME was calculated based on a survey with over 13,000 fisheries experts to assess the effectiveness of fisheries management regimes worldwide in 2008[24]. In addition, we set ecoregions[58] as a random effect to account for the spatial structure of the data. Collinearity among covariates was examined based on bivariate correlations and variance inflation factors (all pairwise correlations were above 0.6 and VIF below 1.5), which led to the exclusion of selected environmental variables (pH, salinity, primary productivity, and minimum sea surface temperature) and social variables (land cover and fisher density), none of which were correlated or collinear with our variables of interest (sustainable-use MPAs and fisheries management effectiveness).

We used the **brms** package[59] to construct the model in the R statistical software. Models were run using the Hamiltonian Monte Carlo algorithm for 10,000 iterations and 4 chains. Posterior estimates were informed by the data alone (weakly informed priors). Convergence was monitored by examining posterior chains for stability and checking if the scale reduction factor was close to 1. Next, we tested a null model with intercepts only and a full model that included all covariates. We compared both models through leave-one-out cross-validation information criteria (LOOIC), ensuring that our full model performed better than the null model (elpd_diff = −95.7). In addition, we used LOOIC to test if the model with interaction performed better than the model without the interaction term between MPA and fisheries management effectiveness (elpd_diff = −0.3). To examine model fit and homoscedasticity, we checked residuals against fitted values and conducted posterior predictive checks (Supplementary Fig. S3). In addition, we evaluated the goodness-of-fit of the model using leave-one-out cross-validation (loo_r2 = 0.41). When predicting biomass in reef polygons, we assumed a model with a random intercept since not all ecoregions with reef polygons are represented in our data.

Biomass predictions per unit area (kg/ha) were then multiplied by the area in each reef polygon to estimate the total reef fish biomass on each reef. Implicitly, we thereby assumed equal biomass density across each reef polygon. Reef polygons range from about 100 m² to 9.8

thousand km², with a median of 6.3 km². We acknowledge that biomass estimates are affected by (i) our reliance on biomass and social and environmental conditions for reefs within our dataset, which may or may not be representative of all reef systems, (ii) potential spatial and temporal imprecision, (iii) other factors not accounted in the model that could also drive biomass, and (iv) social and environmental conditions that can vary over smaller scales than reef polygons considered here.

### Predicting potential changes in catch due to MPA establishment and operation

The potential change in catch from sustainable-use MPAs was estimated by comparing predicted biomass under non-MPA and MPA conditions. To estimate catch from biomass, we used a simple surplus production model[20]. This model assumes that the harvest rate that produces maximum sustainable yield ($F_{MMSY}$) is half of the intrinsic population growth rate ($r$) of the species assemblage. Therefore, assemblages that grow and reproduce faster can sustain higher levels of harvest than slow growing assemblages. A single population-level intrinsic growth rate was assumed for the multispecies assemblage[21] ($r = 0.23$). Catch resulting from this harvest rate will also depend on the standing biomass in each site such that sites with lower biomass should have relatively higher harvest rates than sites with higher biomass (see Supplementary Material for detailed descriptions). As a proxy for $B_{MMSY}$ we used the 90th biomass quantile of predicted biomass for sustainable-use MPA (544 kg/ha), assuming that these sites are fishing at MMSY or multispecies maximum sustainable yield and within limits of globally proposed BMMSY[21,60] (see Supplementary Fig. S4 for a sensitivity analysis). The sensitivity of results to harvest rate assumptions are shown in Supplementary Fig. S5 (growth rates varying from 0.1 to 0.6). Regional patterns and percent changes in catch are not affected by assumptions of harvest rate. However, total absolute numbers of people affected by MPA expansion are sensitive to harvest rate assumptions, with higher harvest rates resulting in more people being benefited. Because the harvest rate is dependent on the population-level intrinsic growth rate, and this can vary significantly according to local conditions, populations with greater abundance of fast-growing species can sustain higher levels of catch and provide greater nutritional benefits.

### Assigning nutritional content to reef fish catch

To assign specific species to the predicted change in reef fish catch, we used the Sea Around Us database[61], allocating total catch estimates to species proportions based on the proportion of reef species caught in each country in 2014. We used SAU to account for country-level differences in catch and because our surveys did not cover all countries containing coral reef polygons. To obtain this information from SAU, we first separated production from artisanal and subsistence sectors. Next, we identified reef species as occurring in the following functional groups: Medium reef assoc. fish (30–89 cm), Large reef assoc. fish (≥90 cm), and Small reef assoc. fish (<30 cm). In addition, we restricted the data to families that were recorded in the underwater visual surveys.

To assign nutritional content to reef fish species, we used the Aquatic Foods Composition Database (AFCD), a comprehensive database containing 3750 records of nutrient content from global databases and peer-reviewed literature[6]. We then use a hierarchical approach[6] to match species-specific taxonomic information with the AFCD and fill nutrient information for species not present in the database. This hierarchy is based on the following order: (1) scientific name, and then the taxa-specific average of (2) genus, (3) family, (4) order, and (5) class. We then matched the median values of the following nutrients: iron, zinc, DHA + EPA, vitamin A, vitamin B$_{12}$, and calcium (Supplementary Fig. S6). These nutrients were chosen because of their high concentration in aquatic species, their importance in

human nutrition, and their inadequate intake across many countries[6]. Because of variability in nutrient values depending on cooking methods and part of the fish used, we used only raw values (excluding cooked, fried, etc) and muscle tissue (excluding bones, head, liver, etc). To avoid potential errors in the data, outlier values for all considered nutrients were checked. We then multiplied the predicted catch by the edible portion of each species based on AFCD data and multiplied further by nutritional value to obtain the total nutrient supply for each nutrient.

### Calculating per capita nutrient supply and catch from MPA expansion

We calculated the per capita nutrient supply by dividing total nutrient supply by the human population around reefs. Although some valuable reef species are traded in international or regional markets, we assumed for simplicity that all catch from MPA expansion will be consumed by coastal communities within a 10 km buffer around reefs (see Supplementary Fig. S7 for sensitivity analysis). Because of this assumption, results should be interpreted as the potential for sustainable-use MPAs to address malnutrition. However, other policies that improve access of the additional catch to vulnerable coastal populations are needed to ensure nutritional benefits. Therefore, at a local scale, the per capita consumption will depend on accessibility: distance of the reef from the community, the size of boats, trade dynamics, as well as other factors such as affordability and dietary preferences. For example, a 10–20 km radius will capture the travel distance that most fishers take in subsistence/artisanal fisheries[62–64]. However, acknowledging the high uncertainty around this value, we tested multiple alternative buffer sizes around reefs to estimate per capita nutrient supply (Supplementary Fig. S7). Larger buffers around reefs increased the number of people impacted, and, thus, lowered per capita nutrient supply. Although the magnitude of impacts changed depending on buffer size, regional patterns were not affected by the assumed human population around reefs.

To calculate the number of people supported by sustainable-use MPAs, we first estimated the per capita reef fish catch by dividing the predicted catch in each reef polygon by the population within a buffer around the reef. Next, we estimated the percent contribution of per capita reef catch relative to per capita national average consumption of aquatic animal-sourced foods based on the Global Nutrient Database (GND)[65]. The GND used the Food and Agriculture Organization of the United Nations Supply and Utilization Accounts to obtain estimates of apparent per capita consumption of 22 food groups and nutrient supply for 156 nutrients across 195 countries. We considered coral reefs to provide a meaningful contribution when predicted coral reef catch represented at least 5% of aquatic animal food intake (see Supplementary Fig. S8 for sensitivity analysis). We assumed the value of at least 5% for two reasons. First, coral reef catch from the species considered in this study represent a median of 3.9% of the total seafood produced in coral reef countries (Supplementary Fig. S18). Second, many coral reefs are close to large populations, which drives per capita consumption down even though not everyone is consuming this catch. Given uncertainty around this assumption, we assumed a range of threshold values to calculate the total number of people nutritionally supported by sustainable-use MPAs. We then summed across all reefs that provided a meaningful contribution to calculate the total number of people that could potentially be supported by existing and future sustainable-use MPAs.

### Calculating the contribution of MPAs to human nutrition

To calculate potential nutritional effects of MPA expansion, we compared a baseline scenario with a scenario of increased reef fish consumption through an expansion of sustainable-use MPAs. Baseline

conditions were calculated using estimates of nutrient consumption in 2017 from the GND. The MPA expansion scenario was calculated by adding the per capita nutrient supply from SUMPA expansion to this baseline level of nutrient intake. Because baseline nutrient consumption is a national-level estimate, coastal communities may have higher reef-fish nutrient intake relative to the national-level average. Therefore, depending on the location, nutritional benefits of sustainable-use MPAs can be underestimated, especially in locations with lower nutrient consumption than the national average.

We then calculated the prevalence of inadequate intake for current conditions and SUMPA expansion scenarios to obtain the difference in inadequate intake across both scenarios. The prevalence of inadequate intake was calculated following three main steps[66]. First, we disaggregated country-level mean intakes into age-sex mean intakes using the Global Expanded Nutrient Supply (GENuS) database for all nutrients except DHA + EPA and vitamin $B_{12}$, which are not included in the GENuS database[67]. Second, using dietary recall data from SPADE (Statistical Program to Assess Habitual Dietary Exposure), we derived habitual dietary intake distributions across age-sex groups and geographies[68]. We used SPADE outputs to describe the shape (gamma or lognormal distribution) of intake distribution for each age-sex group and to derive age-sex mean intakes for DHA + EPA and vitamin $B_{12}$. Lastly, we calculated the prevalence of inadequate intake using the summary exposure values, or SEVs[6,22]. SEVs estimate the population-level risk related to diets by comparing intake distributions with requirements. The latter are continuous risk curves with values of 1 for low intake, 0 for high intakes and 0.5 for intakes at the Estimated Average Requirement (EAR). These absolute risk curves are then constructed as the cumulative normal distribution function of requirements with a mean at the EAR and a coefficient of variation of 10%[69]. EAR estimates were derived from several sources (FAO, Institute of Medicine), and a coefficient of variation of 25% was used to account for uncertainties regarding recommended intakes. For DHA + EPA, we used the relative risk curves that are associated with ischemic heart disease and have different values for adolescent and adult sub-populations (with no risk for children)[22]. The estimated prevalence of inadequate intake range from 0% (no risk) to full population-level risk (100%).

### Propagating uncertainty

We used Monte Carlo simulation to propagate uncertainty across all steps of the analysis. Monte Carlo simulation consists of drawing random numbers from a set of input parameters with known distribution functions to generate a distribution of the output[70]. Therefore, we generated 10,000 model iterations using random values for population growth rate, $B_{MSY}$, coastal population size, and species nutritional value. Values generated followed a normal distribution around the parameter values used in final analysis (see supplementary methods for details on assumed mean and standard deviation values). For each iteration, we calculated the potential number of people nutritionally impacted by expansion of sustainable-use MPAs and thus generated a distribution of results (Supplementary Fig. S9) providing a realistic range of potential impact given uncertainty of parameters.

### Reporting summary

Further information on research design is available in the Nature Portfolio Reporting Summary linked to this article.

## Data availability

The aggregated data generated in this study have been deposited GitHub (https://github.com/danielfvi/SustMPAs-Nutrition)[71]. Reef Life Survey data available online (https://reeflifesurvey.com/). AGRRA data available upon request (https://www.agrra.org/). Aquatic Food Composition Database available through Harvard dataverse[72] (https://dataverse.harvard.edu/dataset.xhtml?persistentId=doi:10.7910/DVN/

KI0NYM). Sea Around Us database is available online (https://www.seaaroundus.org/).

## Code availability

All code used in the analysis is available on GitHub (https://github.com/danielfvi/SustMPAs-Nutrition)[71].

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

## Acknowledgements

We thank the John and Katie Hansen Family Foundation, Betty and Gordon Moore and the National Science Foundation (HNDS 2121239 to C.D.G.) for financial support. C.M.F. was funded through the Arnhold UC Santa Barbara-Conservation International Climate Solutions Collaborative. We thank Atlantic Gulf Rapid Reef Assessment (AGRRA) contributors and data managers, Healthy Reefs Initiative, I. Williams (NOAA Coral Reef Ecosystem Program), NOAA Coral Reef Conservation Program, G. Edgar and R. Stuart-Smith (Reef Life Surveys), Wildlife Conservation Society.

## Author contributions

D.V., C.D.G., M.M., A.Z. and D. Gill conceptualized the research idea, with significant methodological and design input from J.Z.M., A.S., C.M.F., J.S. and N.K. D.V. led paper analysis and visualization, with support from D. Grieco. D.V. drafted the original manuscript, and all co-authors edited and revised the writing.

## Competing interests

M.M. and A.Z. are employees of Conservation International, an organization whose mission is to empower societies to care for nature for the well-being of humanity through science, partnerships, and field demonstrations. The remaining authors declare no competing interests.
