## [Peer Review File · Nature Communications]

Sustainable-use marine protected areas to improve human nutritionReviewers' Comments:

Reviewer #1:

Remarks to the Author:

This article reports the results of a useful analysis quantifying potential nutritional gains from using sustainable-use MPAs as a tool to increase catch, and thus nutrient supply, from coral reef fisheries. While the results are not particularly surprising/novel (i.e., given what we already know about increased biomass/catch potential from MPAs, doing more of them would result in more nutrients) I think it still tends to be useful to have some quantification of these potential effects published so that they can be pointed to in decision-making in the realms of marine conservation and food/nutrition security. I think this article should be published after minor revisions.

Main comments:

The authors should put the quantification of nutritional gains and potential reductions of inadequate intake into context: How can a reader evaluate the magnitude of the number of individuals support by aquatic foods or reducing inadequate nutrient intake? Is this a lot or a little, relatively to, say, current role of aquatic foods or other options to increase nutrient supply? Of course, more nutrients are always better in the context of inadequate supply, but there are multiple options and tradeoffs (and costs/investment) for trying to address inadequate nutrient via different approaches. So, where does the sustainable-use MPA option figure in the broader system, especially given that the figures are estimates for if ALL non-protected reef systems in the world were put into sustainable-use MPAs. Please find a way to contextualize the results.

I also recognize that this analysis draws on existing datasets to estimate impacts at a global level and across many countries, and therefore it is not intended to address feasibility or ground-truth in any particular context. I additionally appreciate attention to various points in the discussion, such as implementation, short-term decreases in nutrient supply, climate change impacts, etc., all of which I had wondered about as reading and then were addressed. However, there are some other methodological decisions that should be justified more explicitly:

1) Why did the authors decide to define 'nutritionally supported' as 5% of aquatic animal source food intake? This seems like potentially a very small amount.

2) Why focus on populations only within 10km of the coast? I understand the need to put the per capita supply into a bit more of a focused context so it is not watered down, and that these are populations who we generally consider should have access to aquatic foods. However, in my experience, in at least some coastal communities around the world, coastal populations already consume high amounts of fish, and it is other foods that they are missing. Also, in other contexts, rural inland populations and urban centers depend heavily on fish (e.g., Bennett et al. 2022). Cambodia strikes me as an example in which it is hard for me to imagine coastal populations not already consuming high amounts of fish (although I could be wrong!). On that note, were there any assumptions in place about the maximum number of servings of fish that an individual would consume over a given period of time, so that you ensure you are not modeling increased per capita fish consumption to an unrealistic extent? Also on the same note, were inadequate nutrient intakes also calculated at this same spatial scale (10k from coast)? I understand them to be derived at the national level based on lines 434-474, while coastal communities may in fact already have higher intake of nutrients found in aquatic foods due to proximity to the coast (as has been shown in the literature). So the calculation of reduction in inadequate intake may be inflated.

Minor comments:

- Panels B and C of Figure 1 are not particularly helpful, as I am interpreting them. I'm not sure what they are adding beyond what is written in the text. Is there supposed to be something on the map panels?

- Line 166: Isn't "availability" a more accurate term than "consumption"?

Refs:

Bennett, A., Rice, E., Muhonda, P., Kaunda, E., Katengeza, S., Liverpool-Tasie, L. S. O., ... & Gondwe, E. (2022). Spatial analysis of aquatic food access can inform nutrition-sensitive policy. *Nature food*, 3(12), 1010-1013.

Reviewer #2:

Remarks to the Author:

Aquatic foods are rich in essential micronutrients and fatty acids, yet their continued provisioning depends on sustainable management of marine and freshwater resources. In this paper, Viana and co-authors bring together disparate data sources and modeling tools from ecosystem assessments, catch modeling, nutrient composition, and nutrient supply estimates to assess what the impact of changing coral reef management from unprotected to sustainable-use Marine Protected Areas (MPAs) would be on inadequate intake of six micronutrients and fatty acids. They find that expanding sustainable use MPAs can indeed prevent cases of inadequate micronutrient intake in many coral reef nations.

I struggled a bit with reviewing this paper. I think the core question is interesting and important – adding a non-conservation motivation for sustainable ocean management – and the combination of datasets and tools is creative and justified, but I found it hard to follow what was done in the methods and didn't find the main body of the paper conveyed the key messages in a convincing way. I would therefore suggest major revisions of this manuscript to organize the materials more clearly and find the right figures and statistics to tell the story of the results.

A few more specific comments and suggestions:

- There are no page and line numbers to use in reviewing this manuscript
- The 2 billion statistic in the first paragraph is outdated, better use the data in this paper instead: <https://pubmed.ncbi.nlm.nih.gov/36240826/>
- In the introduction, there needs to be a bit more information on how different types of MPAs work since this is not an oceans audience
- The maps in Fig. 1 are difficult to compare and don't convincingly show that sustainable use MPAs have higher biomass than non-MPA sites. For Fig 1A, it is unclear how differences in biomass between SUMPAs and non-MPA sites were calculated – by country? Can you say more about these biomass estimate results than just the one paragraph? It seems like a lot of work went into these calculations.
- In the "Expected benefits from..." section, much of the first and second paragraph is methods description and shouldn't be in the results. Parts of it (like the bit about which nutrients were analyzed) could be in the introduction.
- In Fig. 2 I don't follow the terminology of 'potential catch benefits' and 'expected catch effects' when you're talking about 'existing sustainable-use MPAs'. If the MPAs already exist, isn't the catch benefit already realized? Maybe you mean 'estimated'?
- In the 'Expanding sustainable-use MPAs...' section, again the first two paragraphs are mostly methods. Can you focus more on the results?
- In the third paragraph of the 'Expanding sustainable-use MPAs...' section, how are 'nutritionally vulnerable countries' defined?
- In the same paragraph, these are really the core result of the paper to illustrate with some key visuals. Perhaps some bar charts to illustrate ranges for different nutrients / regions / coral reef types / ...
- In Fig. 3F, the percentages are very small (up to 0.5%). Is the unit truly percentage or should this be multiplied by 100?
- In section "Nutritional targeting..." Madagascar is mentioned multiple times – can you add discussion of where sustainable-use MPAs can address multiple nutritional challenges at once?
- In the conclusion, the sentence "implementation of sustainable-use MPAs are usually not part of the food and nutrition agenda of coral reef countries" should be part of the motivation and framing in the introduction. I'm not sure that the reference (Willett et al. 2019) is correct?
- The figure that explains the methods (Fig. M1) is currently in the Supplementary Materials but

should be brought up front, and be expanded with some key numbers on scale of analysis for each of the steps (2518 coral reef sites vs. all coral reef sites vs. by country)

- In the Methods in "Reef fish underwater surveys" the numbers at the end (1117 non-MPA + 804 SUMPAs + 500 no-take) do not add up to 2518 total sites
- In "Predicting fish biomass" what is meant with "ecoregions"?
- It seems that the main findings of the paper come from "Predicting potential changes..." as that is where the differentiation between no MPA and sustainable-use MPAs is done. I am not a fisheries modeling expert but the paper needs more discussion of the implications of the modeling assumptions made here for the final results as they appear to be quite substantial
- There are a couple of other methodological decisions that need discussion of the implications:
 - o Leaving out pelagic fish and invertebrates
 - o Shortcomings of biomass estimates
 - o Assumption that all additional catch will be consumed locally
 - o 5% of aquatic animal foods from coral reef fisheries as a cut-off for 'substantial' contributions seems very low. Depending on what share of total animal foods is from aquatic sources, this could be a very small total contribution to diet. Can you justify why this seems like an appropriate cut-off?
 - o Mixing reef level data with GND (national) and GENU5
 - o Quality of data on fisheries management effectiveness

Reviewer #3:

Remarks to the Author:

Review_Nat Comms

Sustainable-use marine protected areas to improve human nutrition

This paper aims to show the nutritional impacts of increasing the global coverage of sustainable-use marine protected areas. It does this through a series of steps modeling increased production, increased catch, and increased nutritional intake across close to 2000 reef systems globally. This is an important issue for the policy, practice and the scientific communities to engage with. The manuscript was clearly written and methodologically strong. I have two big concerns with the manuscript as written and a few minor comments.

Big

The arc of this paper is based on the implicit assumption that MPAs (in this case SUMPAs) are the policy tool that needs to be used to improve the coastal nutrition inadequacy. While I am sympathetic to that assumption the manuscript needs to set it up as why that is the case AND also be honest about the other approaches. For example, if I were a more traditional fisheries management scientist/economist my first thought would be the this issue is really about Fisheries Management Effectiveness (FME). That is the problem to solve. The authors themselves show that where FME is high there less biomass boost from an MPA. First, it would be nice to see a statistical test around how these distributions differ. Second, how does one go right from the right to the idea that increased FME should not be evaluated as the nutrition intervention. That is an analysis I think that would really benefit fisheries management. E.g. "Improving FME gives us this nutritional boost, but when we add SUMPAs the boost is bigger... but only here, here and here." By not acknowledging that there are these other avenues for nutrition improvements, the manuscript come off as having an agenda... that is eventually proven.

This leads to my second concern if increasing FME can boost nutrition it does so at what cost? This is a very different paper than the ones the authors have written, but it begs the question about the cost of implementing SUMPAs. Why jump right to them? I think there are compelling reasons to do so but this is not addressed. Given the authors cite a really strong paper on the problems and inadequacies of MPA funding it would be good to elevate that part of the discussion. So why SUMPAs? What are the costs? Does the nutrition boost justify those costs cf other ways to increase nutrition? Again, there are

many reasons an SUMPAs is a good idea and I think the authors should lay them out and then say... AND we are looking at the nutrition benefit.

Finally (but related), the argument of the paper is that poor coastal areas near reefs need to eat more fish, so they need to catch more fish, so they need SUMPAs. Aren't they trying to catch more fish? And haven't they been trying to? And is that one of the reasons reefs are depleted? (of course there are other drivers of overfishing ... industrial, invasives, dynamiting etc...). So the SUMPAs need to be placed in places already productive enough for locals but to ensure their longer term sustainability, OR in places that need some rebuilding. If the former then there is no nutrition boost (but there is the mitigation of nutrition loss in the future which is good). If the latter, then COST comes up again. The cost of rebuilding the ecosystem and stocks etc... These costs could be financial, social, but also the loss of off-take, temporary closures, altered off-take zones and timing. The manuscript deals with this problem in one line (Ln 260). This is a significant problem in marine conservation and should be more deeply addressed. This is an especially acute problem for those who "fishing rhymes with poverty" who cannot afford temporary loss of catch.

So, in all, this is a good piece of work and will be an addition to the literature.

Minor Comments

Last sentence of abstract is confusing as written.

Ln 40 – A bit of a slight of hand to go from coastal fisheries to coral reefs. I think it would be better to slow down and explain in two more sentences this relationship/the overlaps/ this importance of the latter to the former.

Ln 61 – "increased production of seafood stocks" ? Is this the best way to characterize this?

Contrast is Figure 1 A, B, C – not existent in my download.

Fig 2. "error lines" ? what are they? Where are they?

Lines 277 on... - Climate Change platitude comes in out of nowhere, is superficial and its import and interaction with the MPA discussion – poorly supported.

Response to Reviewers

Reviewer #1:

This article reports the results of a useful analysis quantifying potential nutritional gains from using sustainable-use MPAs as a tool to increase catch, and thus nutrient supply, from coral reef fisheries. While the results are not particularly surprising/novel (i.e., given what we already know about increased biomass/catch potential from MPAs, doing more of them would result in more nutrients) I think it still tends to be useful to have some quantification of these potential effects published so that they can be pointed to in decision-making in the realms of marine conservation and food/nutrition security. I think this article should be published after minor revisions.

Main comments:

The authors should put the quantification of nutritional gains and potential reductions of inadequate intake into context: How can a reader evaluate the magnitude of the number of individuals support by aquatic foods or reducing inadequate nutrient intake? Is this a lot or a little, relatively to, say, current role of aquatic foods or other options to increase nutrient supply? Of course, more nutrients are always better in the context of inadequate supply, but there are multiple options and tradeoffs (and costs/investment) for trying to address inadequate nutrient via different approaches. So, where does the sustainable-use MPA option figure in the broader system, especially given that the figures are estimates for if ALL non-protected reef systems in the world were put into sustainable-use MPAs. Please find a way to contextualize the results.

Thank you for the great comment. We agree that it is useful to provide more context to the nutritional impacts reported in the manuscript. To illustrate the scale/significance of the potential nutrient contributions, we compared the total reductions in inadequate intake from sustainable-use MPAs to the total reductions from worldwide marine fisheries reform predicted by Golden et al 2021. In this paper, Golden et al estimates a total reduction in inadequate intake from marine wild fisheries reform of 46 million people, representing a ceiling of how much wild fisheries can improve human nutrition. Therefore, nutritional impacts of sustainable-use MPAs represent about 1-7% of the total nutritional impact that reforms in all marine wild fisheries can provide. We added this additional text within the results of the manuscript (lines 210-212):

“Globally, the expansion of sustainable-use MPAs could lead to reductions in inadequate intake across all assessed nutrients for 0.3 - 2.85 million individuals (reduction of 0.2-1.5 million for vitamin B12, 0.1-0.7 million for calcium, 0.05-0.5 million for omega-3 long-chain

polyunsaturated fatty acids (specifically DHA+EPA), 0.07-0.4 million for iron, 0.06-0.2 million for vitamin A, and 0.02-0.15 million for zinc inadequate intakes) (Fig. 4A). Such reduction in inadequate intake represents about 1-7% of the combined total nutritional benefit that reforms in all marine wild capture fisheries could be expected to support.

In addition to providing context to our results, we also reiterate throughout the manuscript that nutritional impacts of sustainable-use MPAs should be viewed as an add-on benefit on top of the main objective of MPAs, which is marine conservation. To reiterate this, we added the following underlined text to:

- The abstract (lines 35-37):

“Coral reef fisheries are a vital source of nutrients for thousands of nutritionally vulnerable coastal communities around the world. Marine protected areas are regions of the ocean designed to preserve or rehabilitate marine ecosystems and thereby increase reef fish biomass. Here, we evaluate the potential effects of expanding a subset of marine protected areas that allow some level of fishing within their borders (sustainable-use MPAs) to improve the nutrition of coastal communities. We estimate that, depending on site characteristics, expanding sustainable-use MPAs could increase catch by up to 20%, which could help prevent 0.3-2.85 million cases of inadequate micronutrient intake in coral reef nations. Our study highlights the potential add-on nutritional benefits of expanding sustainable-use MPAs in coral reef regions and pinpoints locations with the greatest potential to reduce inadequate micronutrient intake level. These findings provide critical knowledge given international momentum to cover 30% of the ocean with MPAs by 2030 and eradicate malnutrition in all its forms.”

- Introduction (lines 73-74):

“Here, we quantified the potential add-on human nutritional benefits that can arise from sustainable-use MPA implementation through increases in local reef fish catch and consumption. We started by compiling information on coral reef fish populations and social and environmental conditions from 2,421 reef sites (1,117 no-take MPA, 804 sustainable-use MPA, 500 non-MPA) in 53 countries to estimate the effect of these areas on standing reef fish biomass. We then used a Bayesian hierarchical model to estimate the expected standing reef fish biomass under non-MPA and sustainable-use MPAs conditions, accounting for other social (e.g., human population, market distance, fisheries governance, human development index) and environmental (e.g., productivity, depth, temperature, wave exposure) variables (see Supplementary Material for details). We then used the model to estimate: 1) the expected biomass and catch for all existing sustainable-use MPAs; 2) the potential biomass and catch associated with an expansion of sustainable-use MPAs to all non-MPA reefs; 3) the change in nutrient supply due to changes in coral reef fish catch; and 4) the potential changes in nutritional inadequacies associated with sustainable-use MPA expansion (see Supplementary Fig. M1 for details). We focused our analyses on zinc, iron, calcium, omega-3 long-chain polyunsaturated fatty acids docosahexaenoic acid

(DHA) and eicosapentaenoic acid (EPA) (hereafter referred as DHA+EPA), and vitamins A and B12, which are abundant in aquatic species and critical for human health."

- Discussion (lines 304-314):

"Nutritional benefits of sustainable-use MPAs should be viewed as an add-on benefit that supplements the marine conservation goals of MPAs. Compared to other measures to address malnutrition, sustainable-use MPAs can be costly and take a long time to provide desired benefits. Other measures such as nutritional supplements and agriculture development can be faster and more cost-effective to address malnutrition⁴⁶. However, sustainable-use MPAs represent a synergistic policy that can meet multiple objectives: conservation, food security, ecosystem resilience, property rights allocation, conflict alleviation, etc., while other targeted policies are usually for single purposes. In addition, sustainable-use MPAs can act as a safety net during food supply shocks, providing a reliable nutrient source easily accessible by local communities. Moreover, sustainable-use MPAs can be important to mitigate future nutritional loss due to anthropogenic actions such as overfishing and climate change⁸."

I also recognize that this analysis draws on existing datasets to estimate impacts at a global level and across many countries, and therefore it is not intended to address feasibility or ground-truth in any particular context. I additionally appreciate attention to various points in the discussion, such as implementation, short-term decreases in nutrient supply, climate change impacts, etc., all of which I had wondered about as reading and then were addressed. However, there are some other methodological decisions that should be justified more explicitly:

1) Why did the authors decide to define 'nutritionally supported' as 5% of aquatic animal source food intake? This seems like potentially a very small amount.

Thank you for the question - this is a great point. The percent contribution of reef catch was calculated relative to national average consumption of aquatic animal sourced foods based on the Global Nutrient Database. We assumed nutritionally supported as **at least** 5% of aquatic animal sourced food for two reasons. First, coral reef catch from the species considered in this study represent a median of 3.9% of the total seafood produced in coral reef countries. Second, many coral reefs are close to large populations, which drives per capita consumption down even though not everyone is consuming this catch. However, we acknowledge the uncertainty around this threshold value and therefore reported the total number of people nutritionally supported by sustainable-use MPAs as a range and not an exact number. We modified the text:

- In the manuscript (lines 150-152) to make it clear that we are assuming a range of threshold values (also see figure S8 for the range of values according to assumptions on the threshold value and buffer around reefs):

“Overlaying MPA boundary data from MPAtlas¹³ with spatial data on reef area²³, we found that 37% of all coral reefs in the world are within sustainable-use MPAs, 11% are within no-take MPAs, and 51% are non-MPA reefs (Supplementary Fig. S1). We predicted the potential human nutritional benefits of expanding sustainable-use MPAs to all non-MPA reefs (~70,000 km², Fig. 3A), by (i) calculating the potential increase in biomass and thus catch from sustainable-use MPA expansion, (ii) estimating the total number of people that could be nutritionally supported by existing and future sustainable-use MPAs, and (iii) calculating the potential change in the risk of prevalence of inadequate nutrient intake with sustainable-use MPA expansion (see Supplementary Fig. M1 in supplementary methods for details). We defined nutritional support from sustainable-use MPAs as the provision of at least 5% of aquatic animal source food intake from coral reef catch (see Supplementary Fig. S8 for sensitivity analysis). We then summed the population near reefs (10km) above a range of threshold (5-30%) to estimate the total number of people supported by sustainable-use MPAs. We found that sustainable-use MPAs could support the nutritional needs of 2.8 to 30 million people by substantially contributing to overall aquatic animal-source food intake.”

- Added additional information in the methods around this assumption (lines 494-500):

“We considered coral reefs to provide a meaningful contribution when predicted coral reef catch represented at least 5% of aquatic animal food intake (see Supplementary Fig. S8 for sensitivity analysis). We assumed the value of at least 5% for two reasons. First, coral reef catch from the species considered in this study represent a median of 3.9% of the total seafood produced in coral reef countries (Supplementary Fig. S18). Second, many coral reefs are close to large populations, which drives per capita consumption down even though not everyone is consuming this catch. Given uncertainty around this assumption, we assumed a range of threshold values to calculate the total number of people nutritionally supported by sustainable-use MPAs. We then summed across all reefs that provided a meaningful contribution to calculate the total number of people that could potentially be supported by existing and future sustainable-use MPAs.”

In addition, we included the figure below in the supplementary material.

Figure S18 - Contribution of reef fish to national aquatic food production within coral reef countries.

2) Why focus on populations only within 10km of the coast? I understand the need to put the per capita supply into a bit more of a focused context so it is not watered down, and that these are populations who we generally consider should have access to aquatic foods. However, in my experience, in at least some coastal communities around the world, coastal populations already consume high amounts of fish, and it is other foods that they are missing. Also, in other contexts, rural inland populations and urban centers depend heavily on fish (e.g., Bennett et al. 2022). Cambodia strikes me as an example in which it is hard for me to imagine coastal populations not already consuming high amounts of fish (although I could be wrong!). On that note, were there any assumptions in place about the maximum number of servings of fish that an individual would consume over a given period of time, so that you ensure you are not modeling increased per capita fish consumption to an unrealistic extent? Also on the same note, were inadequate nutrient intakes also calculated at this same spatial scale (10k from coast)? I understand them to be derived at the national level based on lines 434-474, while coastal communities may in fact already have higher intake of nutrients found in aquatic foods due to proximity to the coast (as has been shown in the literature). So the calculation of reduction in inadequate intake may be inflated.

Thank you for the great comment.

We selected a 10 km buffer based on the evidence from the literature that estimates the distance artisanal fishers travel to fishing grounds (e.g. Clark et al 2002, Dunn et al 2010).

However, we recognize that our results are sensitive to this choice, so we evaluated the sensitivity of our results to alternative buffer sizes ranging from 5-30 km. Larger buffers around reefs increased the number of people impacted and lowered the incremental additions to per capita nutrient supply. Although the magnitude of impacts changed depending on buffer size, regional patterns were not affected by the assumed population around reefs. Discussion around this assumption can be found in methods (lines 473-481):

“We calculated the per capita nutrient supply by dividing total nutrient supply by the human population around reefs. Although some valuable reef species are traded in international or regional markets, we assumed for simplicity that all catch from MPA expansion will be consumed by coastal communities within a 10 km buffer around reefs (see Supplementary Fig. S7 for sensitivity analysis). Because of this assumption, results should be interpreted as the potential for sustainable-use MPAs to address malnutrition. However, other policies that improve access of the additional catch to vulnerable coastal populations are needed to ensure nutritional benefits. Therefore, at a local scale, the per capita consumption will depend on accessibility: distance of the reef from the community, the size of boats, trade dynamics, as well as other factors such as affordability and dietary preferences. For example, a 10-20 km radius will capture the travel distance that most fishers take in subsistence/artisanal fisheries⁶²⁻⁶⁴. However, acknowledging the high uncertainty around this value, we tested multiple alternative buffer sizes around reefs to estimate per capita nutrient supply (Supplementary Fig. S7). Larger buffers around reefs increased the number of people impacted, and, thus, lowered per capita nutrient supply. Although the magnitude of impacts changed depending on buffer size, regional patterns were not affected by the assumed human population around reefs.”

Regarding high consumption of fish in coastal communities - this is also important to understand. Although this can be true in some places, several coral reef systems around the globe are experiencing heavy overfishing, which diminishes the availability of fish for local consumption. Because of this, many communities may not have enough fish to supply the entire population, given that not everyone living close to reefs are fishers. For communities that already consume large amounts of fish, the main role of sustainable-use MPAs is to *maintain* availability of seafood for local consumption, preventing future cases of inadequate intake. To investigate this further, we compared original fish intakes (from Global Nutrient Database) versus estimated intakes with sustainable-use MPAs (see figure S17 below). As you can see in the figure below, the added catch is small relative to original intakes within each country. This happens because the added catch from sustainable-use MPAs is distributed across the entire coastal population around the MPAs.

We added additional text to the discussion to clarify this point:

- Lines 264-266:

“Many factors determine the magnitude of impacts on a local scale. These factors are related to the availability, access, and utilization of reef fish caught from sustainable-use MPAs. For example, our analysis suggests that availability will be influenced by local factors such as the productivity of the local reef system, the number of seafood consumers (which impacts per capita consumption), the prevalence of inadequate intake, and the efficacy of local fisheries management. However, for additional catch through sustainable-use MPA expansion to have an impact on nutritional health, it is important that those in need of nutrients have access to increased catch from sustainable-use MPAs. In cases where communities already have high consumption of fish³³, sustainable-use MPAs can be key to maintain availability and access to nutrients from coral reefs. With seafood being one of the most widely traded food commodities in the world^{34,35}, it is important to ensure that SUMPAs catch is affordable and accessible for domestic consumption³⁶. In addition, cultural and dietary preferences and fish utilization practices can also influence nutritional impacts of SUMPAs. Factors such as sanitary conditions of the fish caught and sold, and seafood loss and waste can also fundamentally influence shelf life, nutritional quality, and availability of reef fish³⁷”

- Additionally, we added this figure to the supplementary material (Figure S17):

Figure S17 - Comparison of original intakes versus estimated intakes with implementation of sustainable-use MPAs

Minor comments:

- Panels B and C of Figure 1 are not particularly helpful, as I am interpreting them. I'm not sure what they are adding beyond what is written in the text. Is there supposed to be something on the map panels?

Panels B and C show the distribution of the sample sites and their biomass estimates. We think it is important to have these maps to show the distribution of our data, which has greater coverage in Australia and the Caribbean (especially for sustainable-use MPA sites). In addition, it helps readers understand how underwater survey data was used in the analysis. To improve interpretation of this figure, we changed the order of the panels within the figure.

- Line 166: Isn't "availability" a more accurate term than "consumption"?

Great suggestion. We changed "consumption" to "availability".

Refs:

Bennett, A., Rice, E., Muhonda, P., Kaunda, E., Katengeza, S., Liverpool-Tasie, L. S. O., ... & Gondwe, E. (2022). Spatial analysis of aquatic food access can inform nutrition-sensitive policy. *Nature food*, 3(12), 1010-1013.

Reviewer #2:

Aquatic foods are rich in essential micronutrients and fatty acids, yet their continued provisioning depends on sustainable management of marine and freshwater resources. In this paper, Viana and co-authors bring together disparate data sources and modeling tools from ecosystem assessments, catch modeling, nutrient composition, and nutrient supply estimates to assess what the impact of changing coral reef management from unprotected to sustainable-use Marine Protected Areas (MPAs) would be on inadequate intake of six micronutrients and fatty acids. They find that expanding sustainable use MPAs can indeed prevent cases of inadequate micronutrient intake in many coral reef nations.

I struggled a bit with reviewing this paper. I think the core question is interesting and important – adding a non-conservation motivation for sustainable ocean management – and the combination of datasets and tools is creative and justified, but I found it hard to follow what was done in the methods and didn't find the main body of the paper conveyed the key messages in a convincing way. I would therefore suggest major revisions of this manuscript to organize the materials more clearly and find the right figures and statistics to tell the story of the results.

Thank you for the comment. We have addressed this comment in several ways throughout the manuscript that clarified the methods and improved how key messages are conveyed.

To clarify the methods, we referenced the methodological framework figure in the supplementary material throughout the manuscript and added further discussion to key assumptions of the analysis, such as:

- Limitations of underwater fish survey data. Although pelagic species and invertebrates can be important for fisheries and nutrient supply of coastal populations, we were unable to include this in the analysis because of data restrictions. We used underwater surveys as the primary source of data for the model and this method do not accurately capture biomass of pelagic fish and focuses on reef fish only (excluding invertebrates). Based on our data, pelagic and transient fish represented a median of 15% of the total biomass across all sample sites. We acknowledge that because of this limitation, the results presented here should be considered a conservative estimate of the total nutritional value of coral reefs. In addition, there are inherent limitations to underwater surveys. For example, underwater visual surveys can be affected by the behavior of different fish species and size of larger fish tend to be underestimated. To address this, we revised this section of the methods (lines 345-353):

“Because underwater fish counts do not accurately capture biomass of large schools of pelagic fish (e.g., Scombrids, Sphyraenids) or large transient fish (including shark and ray species), we removed them from the analysis^{52,53}. Depending on the site, pelagic and large transient fish can be important for biomass, catch, and nutrition. Based on the fish captured in the underwater surveys, pelagic and transient fish represented a median of 15% of the total biomass across all sample sites. In addition, because of data constraints, we only considered reef fish catch as a nutrient source, however, invertebrate species and aquatic plants can also be an important source of nutrients in many low-income countries. Finally, underwater visual surveys can be affected by the behavior of different fish species and size of larger fish tend to be underestimated⁵³. Therefore, results presented here are likely an underestimate of the total nutritional benefits that sustainable-use MPAs can provide to local populations.”

- Threshold for meaningful contribution of reef fish. We assumed nutritionally supported as **at least** 5% of aquatic animal sourced food for two reasons. First, coral reef catch from the species considered in this study represent a median of 3.9% of the total seafood produced in coral reef countries. Second, many coral reefs are close to large populations, which drives per capita consumption down even though not everyone is consuming this catch. However, we acknowledge the uncertainty around this threshold value and therefore reported the total number as a range and

not an exact number. We added further text to the methods to clarify this assumption (lines 494-500):

“We considered coral reefs to provide a meaningful contribution when predicted coral reef catch represented at least 5% of aquatic animal food intake (see Supplementary Fig. S8 for sensitivity analysis). We assumed the value of at least 5% for two reasons. First, coral reef catch from the species considered in this study represent a median of 3.9% of the total seafood produced in coral reef countries (Supplementary Fig. S18). Second, many coral reefs are close to large populations, which drives per capita consumption down even though not everyone is consuming this catch. Given uncertainty around this assumption, we assumed a range of threshold values to calculate the total number of people nutritionally supported by sustainable-use MPAs. We then summed across all reefs that provided a meaningful contribution to calculate the total number of people that could potentially be supported by existing and future sustainable-use MPAs.”

- Assumption that all additional catch will be consumed locally. We acknowledge that not all the catch from sustainable-use MPAs are consumed locally. The portion of the catch that is locally consumed depends on local factors such as market access, value of species captured, local preferences, etc. Because this is a global model and local and market dynamics vary greatly depending on the region, we assumed for simplicity that all catch from MPA expansion will be consumed by coastal communities. To further clarify the motivation and implications of this assumption, we added additional text to the methods (lines 473-481):

“We calculated the per capita nutrient supply by dividing total nutrient supply by the human population around reefs. Although some valuable reef species are traded in international or regional markets, we assumed for simplicity that all catch from MPA expansion will be consumed by coastal communities within a 10 km buffer around reefs (see Supplementary Fig. S7 for sensitivity analysis). Because of this assumption, results should be interpreted as the potential for sustainable-use MPAs to address malnutrition. However, other policies that improve access of the additional catch to vulnerable coastal populations are needed to ensure nutritional benefits. Therefore, at a local scale, the per capita consumption will depend on accessibility: distance of the reef from the community, the size of boats, trade dynamics, as well as other factors such as affordability and dietary preferences. For example, a 10-20 km radius will capture the travel distance that most fishers take in subsistence/artisanal fisheries⁶²⁻⁶⁴. However, acknowledging the high uncertainty around this value, we tested multiple alternative buffer sizes around reefs to estimate per capita nutrient supply (Supplementary Fig. S7). Larger buffers around reefs increased the number of people impacted, and, thus, lowered per capita nutrient supply.

Although the magnitude of impacts changed depending on buffer size, regional patterns were not affected by the assumed human population around reefs. ”

To improve how key messages are conveyed we updated Fig 4 to also show the potential nutritional impacts of SUMPAs globally, across different nutrients and across different regions of the world. This was a great addition to the manuscript. We appreciate your comment, which helped us to significantly improve the clarity of our methodological approach and to convey our main results.

Original figure was:

Updated figure:

In addition, we added additional text to the results highlighting results across regions (lines 212–218):

“Globally, the expansion of sustainable-use MPAs could lead to reductions in inadequate intake across all assessed nutrients for 0.3 - 2.85 million individuals (reduction of 0.2-1.5 million for vitamin B₁₂, 0.1-0.7 million for calcium, 0.05-0.5 million for omega-3 long-chain polyunsaturated fatty acids (specifically DHA+EPA), 0.07-0.4 million for iron, 0.06-0.2 million for vitamin A, and 0.02-0.15 million for zinc inadequate intakes) (Fig. 4A). Such reduction in inadequate intake represents about 1-7% of the combined total nutritional benefit that reforms in all marine wild capture fisheries could be expected to support⁶. Regions such as South and Southeast Asia, Pacific, Latin America and Caribbean (see Supplementary Table 2 for a complete list of countries) have the highest potential for nutritional improvement following implementation of effective sustainable-use MPAs (Fig 4A). For other regions such as Sub-Saharan Africa, sustainable-use MPAs can also have important effects at a local level, but on aggregate at national scales, total impact is relatively low due to reduced total reef area and lower numbers of people living around reefs.”

A few more specific comments and suggestions:

- There are no page and line numbers to use in reviewing this manuscript

We apologize for this oversight in the original submission. We added page and line numbers to the revised submission.

- The 2 billion statistic in the first paragraph is outdated, better use the data in this paper instead: <https://pubmed.ncbi.nlm.nih.gov/36240826/>

We appreciate this recommendation but Stevens et al. (2022) only focus on preschool-aged children and women of reproductive age only. Although it is a reliable source for these population groups, the statistics from the FAO 2020 analysis covers all population groups. We therefore opted to maintain the 2 billion figure. We added a reference to Stevens et al (2022) later in the manuscript.

- In the introduction, there needs to be a bit more information on how different types of MPAs work since this is not an oceans audience

Thank you for the suggestion. We added additional text to the second paragraph of the introduction to clarify the differences between “no-take” and “sustainable-use” MPAs (lines 55-56).

“Marine protected areas (MPAs) are increasingly used to protect coral reef ecosystems and recover associated fisheries from depletion globally⁹. MPAs are areas of the ocean with specific rules and restrictions primarily designed to protect marine ecosystems from anthropogenic threats^{10,11}. MPAs can be broadly categorized into “no-take” areas where all fishing is strictly prohibited and “partially protected” or “sustainable-use” areas where some forms of fishing are permitted. Sustainable-use MPAs use various fisheries management tools to support fisheries sustainability and may be zoned for multiple uses (e.g., tourism, aquaculture, conservation). Both MPA types generally have significant positive effects on the biomass of targeted fish within their boundaries compared to neighboring non-MPA areas¹². Currently, there are about seventeen thousand MPAs worldwide, covering about 8.2% of the ocean¹³. Despite failing to reach the globally agreed target of 10% MPA coverage by 2020¹⁴, there is a new global commitment to cover 30% of our oceans with MPAs and other effective area-based conservation measures (OECMs) by 2030^{14–16}.”

- The maps in Fig. 1 are difficult to compare and don't convincingly show that sustainable use MPAs have higher biomass than non-MPA sites. For Fig 1A, it is unclear how differences in biomass between SUMPAs and non-MPA sites were calculated – by country? Can you say more about these biomass estimate results than just the one paragraph? It seems like a lot of work went into these calculations.

Thank you for the question and suggestion to add more information to this section. Differences in biomass are the estimated effect sizes of sustainable-use MPAs relative to

non-MPA sites. We have added additional text to clarify how differences in biomass were calculated (lines 97-98):

“We estimated the potential net conservation benefits of sustainable-use MPA establishment by examining the effect of sustainable-use MPAs on reef fish biomass (Fig. 1A). Biomass estimates are based on underwater surveys from around the world, with most observations from Australia, the Caribbean and Southwest Pacific (Fig. 1A and B). Globally, we found that sustainable-use MPAs have on average 15% more biomass than non-MPA sites, although differences were highly dependent on the effectiveness of fisheries management in surrounding waters (Fig. 1C). Estimated differences in biomass were calculated based on the effect size of sustainable-use MPAs on reef fish biomass. Locations with high fisheries management effectiveness (FME; Fig. 1C) had a lower difference in biomass between non-MPA and sustainable-use MPA biomass relative to locations with low management effectiveness, likely because non-MPA sites in high FME locations are already well-managed and therefore less depleted.”

Additionally, we have added additional text to the figure caption to further clarify how differences in biomass were calculated.

Fig. 1. Biomass in sustainable-use marine protected areas (MPAs), highlighting A) distribution of percent differences in biomass of sustainable-use MPAs compared to non-MPA sites for locations with high (>0.5) and low (≤ 0.5) estimated fisheries management effectiveness (FME). Differences in biomass are the estimated effect size of sustainable-use MPAs relative to non-MPA sites. Maps show B) observed biomass in non-MPA sites (n=1,117), and C) observed biomass in sustainable-use MPA sites (n=804). The distributions of both values are indicated as box plots on the left-hand side of panels A and B.

To better discuss biomass estimate results, we added additional text related to the distribution of underwater survey sites (lines 92-94):

“We estimated the potential net conservation benefits of sustainable-use MPA establishment by examining the effect of sustainable-use MPAs on reef fish biomass (Fig. 1A). Biomass estimates are based on underwater surveys from around the world, with most observations from Australia, the Caribbean and Southwest Pacific (Fig. 1A and B). Globally, we found that sustainable-use MPAs have on average 15% more biomass than non-MPA sites, although differences were highly dependent on the effectiveness of fisheries management in surrounding waters (Fig. 1C). Estimated differences in biomass were calculated based on the effect size of sustainable-use MPAs on reef fish biomass. Locations with high fisheries management effectiveness (FME; Fig. 1C) had a lower difference in

biomass between non-MPA and sustainable-use MPA biomass relative to locations with low management effectiveness, likely because non-MPA sites in high FME locations are already well-managed and therefore less depleted.”

- In the “Expected benefits from...” section, much of the first and second paragraph is methods description and shouldn’t be in the results. Parts of it (like the bit about which nutrients were analyzed) could be in the introduction.

Thank you for the suggestion. We acknowledge that there are many methods included in the main text. However, because this is a “methods last” format journal, it seems critical and common practice to include a brief outline of the analytical approach into the main text to maintain clarity for the reader. Nonetheless, we have removed some of the methods text (e.g. lines 127, 158) and moved the part about the nutrients analyzed to the introduction (lines 86-89):

“Here, we quantified the potential add-on human nutritional benefits that can arise from sustainable-use MPA implementation through increases in local reef fish catch and consumption. We started by compiling information on coral reef fish populations and social and environmental conditions from 2,421 reef sites (1,117 no-take MPA, 804 sustainable-use MPA, 500 non-MPA) in 53 countries to estimate the effect of these areas on standing reef fish biomass. We then used a Bayesian hierarchical model to estimate the expected standing reef fish biomass under non-MPA and sustainable-use MPAs conditions, accounting for other social (e.g., human population, market distance, fisheries governance, human development index) and environmental (e.g., productivity, depth, temperature, wave exposure) variables (see Supplementary Material for details). We then used the model to estimate: 1) the expected biomass and catch for all existing sustainable-use MPAs; 2) the potential biomass and catch associated with an expansion of sustainable-use MPAs to all non-MPA reefs; 3) the change in nutrient supply due to changes in coral reef fish catch; and 4) the potential changes in nutritional inadequacies associated with sustainable-use MPA expansion (see Supplementary Fig. M1 for details). We focused our analyses on zinc, iron, calcium, omega-3 long-chain polyunsaturated fatty acids docosahexaenoic acid (DHA) and eicosapentaenoic acid (EPA) (hereafter referred as DHA+EPA), and vitamins A and B₁₂, which are abundant in aquatic species and critical for human health⁶.”

- In Fig. 2 I don’t follow the terminology of ‘potential catch benefits’ and ‘expected catch effects’ when you’re talking about ‘existing sustainable-use MPAs’. If the MPAs already exist, isn’t the catch benefit already realized? Maybe you mean ‘estimated’?

Thank you for the suggestion. It is correct that, in principle, these benefits have already been realized, but they are not directly known; therefore, they must be estimated.

Reframing these catch benefits as “estimated” has improved clarity. We have changed the terminology of figure 2 to “estimated catch benefits” and “estimated catch effects” in the figure caption and changed the y-axis title of the figure from “Expected difference in catch” to “Estimated difference in catch”.

- In the ‘Expanding sustainable-use MPAs...’ section, again the first two paragraphs are mostly methods. Can you focus more on the results?

Thank you for the suggestion. As stated before, because this is a “methods last” format journal, we think it is important to include a minimum description of the methods into the main text to maintain clarity for the reader. We have reduced some of the text related to methods to focus more on the results.

- In the third paragraph of the ‘Expanding sustainable-use MPAs...’ section, how are ‘nutritionally vulnerable countries’ defined?

We added the following underlined text to better emphasize the definition we adopted for identifying nutritionally vulnerable countries: “vulnerable countries (i.e., populations with inadequate intakes higher than 25% across all selected nutrients)” (lines 163-164).

- In the same paragraph, these are really the core result of the paper to illustrate with some key visuals. Perhaps some bar charts to illustrate ranges for different nutrients / regions / coral reef types / ...

We updated Figure 4 to better illustrate these ranges by country type (all vs. vulnerable), region, and nutrient (see original and updated figures above). In addition, we added text in the results highlighting impact across regions and nutrients (see original and updated figure above).

- In Fig. 3F, the percentages are very small (up to 0.5%). Is the unit truly percentage or should this be multiplied by 100?

Thank you for the comment. Yes - these are percentages. Most countries have very low values of changes in SEV, but even small changes can add up to large numbers of people as this is the percent risk of nutritional inadequacies over the entire population. Also, several countries have SEV larger than 0.5%, but to show patterns in countries with low values, we grouped all countries with SEV larger than 0.5%.

- In section “Nutritional targeting...” Madagascar is mentioned multiple times – can you add discussion of where sustainable-use MPAs can address multiple nutritional challenges at once?

Thank you for the suggestion. We added discussion on specific nutrients to highlight how each nutrient supports human health. We also changed the examples to make sure no country is repeated in the text so as to diversify the results and broaden relevance (line 223):

“Because each country has different specific nutrient inadequacies, sustainable-MPAs can be strategically placed in areas with the highest potential to reduce specific nutritional inadequacies (Fig. 4). For example, creating targeted sustainable-use MPAs in Yemen and Kenya has the potential to reduce inadequate intake risks of omega-3 long-chain polyunsaturated fatty acids (DHA+EPA). Increased intake of DHA+EPA can promote brain and eye health and is associated with reduced risk of heart disease²⁵. India and Bangladesh could particularly benefit from increased supply of vitamin B₁₂, where inadequacy is more than twice the global average⁶. Vitamin B₁₂ deficiency is associated with increased risk of heart disease and cognitive decline²⁶. Mozambique and Cambodia could benefit from increased supply of iron, which is particularly important for healthy brain development and growth in children²⁷, and can prevent maternal mortality²⁵. Sustainable-use MPAs in Nicaragua and Madagascar have the potential to reduce inadequate intake of zinc, which supports immunity and is particularly important for children and pregnant women²⁸. Kuwait and Indonesia could benefit mostly from increased supply of calcium, which supports bone health and blood pressure²⁹. Lastly, Oman and Kiribati could benefit from increased vitamin A supply, which supports eye health and cell growth³⁰.”

- In the conclusion, the sentence “implementation of sustainable-use MPAs are usually not part of the food and nutrition agenda of coral reef countries” should be part of the motivation and framing in the introduction. I’m not sure that the reference (Willett et al. 2019) is correct?

Thank you for the comment. We have deleted this sentence from the discussion paragraph and added a new paragraph to the introduction for this to be part of the motivation (lines 63-64):

“The integration of sustainable-use MPAs into the food and nutrition agendas of coral reef countries is often overlooked¹⁷. While good fisheries management holds the potential to enhance human nutrition⁶, its success is contingent upon the availability of high-quality data and robust management capacities. Sustainable-use MPAs emerge as a realistic and cost-effective intervention in various settings, offering a balance between conservation

goals and sustainable resource utilization¹⁸. Moreover, these conservation measures play a crucial role in safeguarding the rights of local communities, protecting them from exploitation by non-local entities¹⁹. This not only underscores the significance of MPAs as a pivotal tool in sustainable resource management but also highlights their potential multifaceted impacts on both ecological conservation and human nutrition.”

- The figure that explains the methods (Fig. M1) is currently in the Supplementary Materials but should be brought up front, and be expanded with some key numbers on scale of analysis for each of the steps (2518 coral reef sites vs. all coral reef sites vs. by country)

Thank you for the suggestion. We have updated the figure to include these suggestions. Although we agree that this figure is key to understanding the analysis, we think that it would take up too much space in the main manuscript. We added references to better highlight this figure throughout the manuscript (line 86, 148).

- In the Methods in “Reef fish underwater surveys” the numbers at the end (1117 non-MPA + 804 SUMPA + 500 no-take) do not add up to 2518 total sites

Thank you for catching this error. We corrected the total number of sites (2,421).

- In “Predicting fish biomass” what is meant with “ecoregions”?

Thank you for pointing this out. We have added a reference for ecoregions (line 392) and added a definition in the supplementary material:

“Ecoregions: Marine Ecoregions of the World (MEOW) classification system that defines 232 marine ecoregions. The MEOW system is a biogeographic classification of the world's coasts and shelves (Spalding et al 2007).”

- It seems that the main findings of the paper come from “Predicting potential changes...” as that is where the differentiation between no MPA and sustainable-use MPAs is done. I am not a fisheries modeling expert but the paper needs more discussion of the implications of the modeling assumptions made here for the final results as they appear to be quite substantial

Thank you for the comment. These are the demographic parameters on which our results are most dependent. We expanded our discussion on the implications of assumptions around growth and harvest rate to the overall results (lines 439-442):

“The potential change in catch from sustainable-use MPAs was estimated by comparing predicted biomass under non-MPA and MPA conditions. To estimate catch from biomass, we used a simple surplus production model²⁰. This model assumes that the harvest rate that produces maximum sustainable yield (F_{MMSY}) is half of the intrinsic population growth rate (r) of the species assemblage. Therefore, assemblages that grow and reproduce faster can sustain higher levels of harvest than slow growing assemblages. A single population-level intrinsic growth rate was assumed for the multispecies assemblage²¹ ($r = 0.23$). Catch resulting from this harvest rate will also depend on the standing biomass in each site such that sites with lower biomass should have relatively higher harvest rates than sites with higher biomass (see Supplementary Material for detailed descriptions). As a proxy for B_{MMSY} we used the 90th biomass quantile of predicted biomass for sustainable-use MPA (544 kg/ha), assuming that these sites are fishing at MMSY or multispecies maximum sustainable yield and within limits of globally proposed B_{MMSY} ^{21,60} (see Supplementary Fig. S4 for a sensitivity analysis). Sensitivity of results to harvest rate assumptions are shown in Supplementary Fig. S5 (growth rates varying from 0.1 to 0.6). Regional patterns and percent changes in catch are not affected by assumptions of harvest rate. However, total absolute numbers of people affected by MPA expansion are sensitive to harvest rate assumptions, with higher harvest rates resulting in more people being benefited. Because harvest rate is dependent on the population-level intrinsic growth rate, and this can vary significantly according to local conditions, populations with greater abundance of fast-growing species can sustain higher levels of catch and provide greater nutritional benefits.”

- There are a couple of other methodological decisions that need discussion of the implications:
 - o Leaving out pelagic fish and invertebrates

Thank you for the comment. Based on the fish captured in the underwater surveys, we estimated that pelagic and transient fish represented a median of 15% of the total biomass across all sample sites. We have expanded the discussion around this assumption on lines 345-347:

“We compiled information of coral reef transects from the Reef Life Survey (RLS)⁴⁹ and Atlantic and Gulf Rapid Reef Assessment (AGRRA) databases⁵⁰. Both databases are based on underwater fish counts by size class within a belt transect conducted from 1997 to 2020. We then calculated individual biomass by using length-weight relationships published for all species on Fishbase⁵¹ and then multiplying individual biomass by the total number of fish within each size class. The final compiled database contained 16,365 surveys from 2,421 tropical coral reef sites (i.e., within 23.5 latitude degrees) distributed across 53 countries. Where data from multiple years were available for a single site, we included only the most recent year. To estimate the “fishable biomass” we retained only

fish larger than 10 cm⁵². Because underwater fish counts do not accurately capture biomass of large schools of pelagic fish (e.g., Scombrids, Sphyraenids) or large transient fish (including shark and ray species), we removed them from the analysis^{52,53}. Depending on the site, pelagic and large transient fish can be important for biomass, catch, and nutrition. Based on the fish captured in the underwater surveys, pelagic and transient fish represented a median of 15% of the total biomass across all sample sites. In addition, because of data constraints, we only considered reef fish catch as a nutrient source, however, invertebrate species and aquatic plants can also be an important source of nutrients in many low-income countries. Finally, underwater visual surveys can be affected by the behavior of different fish species and size of larger fish tend to be underestimated⁵³. Therefore, results presented here are likely an underestimate of the total nutritional benefits that sustainable-use MPAs can provide to local populations.”

o Shortcomings of biomass estimates

Thank you for the comment. We added additional text to the methods explaining bias related to biomass estimates from underwater visual surveys (lines 350-353):

“We compiled information of coral reef transects from the Reef Life Survey (RLS)⁴⁹ and Atlantic and Gulf Rapid Reef Assessment (AGRRA) databases⁵⁰. Both databases are based on underwater fish counts by size class within a belt transect conducted from 1997 to 2020. We then calculated individual biomass by using length-weight relationships published for all species on Fishbase⁵¹ and then multiplying individual biomass by the total number of fish within each size class. The final compiled database contained 16,365 surveys from 2,421 tropical coral reef sites (i.e., within 23.5 latitude degrees) distributed across 53 countries. Where data from multiple years were available for a single site, we included only the most recent year. To estimate the “fishable biomass” we retained only fish larger than 10 cm⁵². Because underwater fish counts do not accurately capture biomass of large schools of pelagic fish (e.g., Scombrids, Sphyraenids) or large transient fish (including shark and ray species), we removed them from the analysis^{52,53}. Depending on the site, pelagic and large transient fish can be important for biomass, catch, and nutrition. Based on the fish captured in the underwater surveys, pelagic and transient fish represented a median of 15% of the total biomass across all sample sites. In addition, because of data constraints, we only considered reef fish catch as a nutrient source, however, invertebrate species and aquatic plants can also be an important source of nutrients in many low-income countries. Finally, underwater visual surveys can be affected by the behavior of different fish species and size of larger fish tend to be underestimated⁵³. Therefore, results presented here are likely an underestimate of the total nutritional benefits that sustainable-use MPAs can provide to local populations.”

o Assumption that all additional catch will be consumed locally

Thank you for the suggestions. We added additional discussion around this assumption in lines 473-478:

“We calculated the per capita nutrient supply by dividing total nutrient supply by the human population around reefs. Although some valuable reef species are traded in international or regional markets, we assumed for simplicity that all catch from MPA expansion will be consumed by coastal communities within a 10 km buffer around reefs (see Supplementary Fig. S7 for sensitivity analysis). Because of this assumption, results should be interpreted as the potential for sustainable-use MPAs to address malnutrition. However, other policies that improve access of the additional catch to vulnerable coastal populations are needed to ensure nutritional benefits. Therefore, at a local scale, the per capita consumption will depend on accessibility: distance of the reef from the community, the size of boats, trade dynamics, as well as other factors such as affordability and dietary preferences. For example, a 10-20 km radius will capture the travel distance that most fishers take in subsistence/artisanal fisheries⁶²⁻⁶⁴. However, acknowledging the high uncertainty around this value, we tested multiple alternative buffer sizes around reefs to estimate per capita nutrient supply (Supplementary Fig. S7). Larger buffers around reefs increased the number of people impacted, and, thus, lowered per capita nutrient supply. Although the magnitude of impacts changed depending on buffer size, regional patterns were not affected by the assumed human population around reefs.”

o 5% of aquatic animal foods from coral reef fisheries as a cut-off for ‘substantial’ contributions seems very low. Depending on what share of total animal foods is from aquatic sources, this could be a very small total contribution to diet. Can you justify why this seems like an appropriate cut-off?

Thank you for the question - this is a great point that we agree needs further clarification. Please see the response above related to this assumption.

o Mixing reef level data with GND (national) and GENU S

Thank you for the comment. We added additional discussion related to this assumption in lines 509-512 of the methods:

“To calculate potential nutritional effects of MPA expansion, we compared a baseline scenario with a scenario of increased reef fish consumption through an expansion of sustainable-use MPAs. Baseline conditions were calculated using estimates of nutrient consumption in 2017 from the GND. The MPA expansion scenario was calculated by adding

the per capita nutrient supply from SUMPA expansion to this baseline level of nutrient intake. Because baseline nutrient consumption is a national-level estimate, coastal communities may have higher reef-fish nutrient intake relative to the national-level average. Therefore, depending on the location, nutritional benefits of sustainable-use MPAs can be underestimated, especially in locations with lower nutrient consumption than the national average.

o Quality of data on fisheries management effectiveness

Thank you for the comment. We added additional information about how this parameter was calculated in the methods (lines 390-391):

“We used a two-level linear Bayesian model with a normal error structure to predict log reef fish biomass (above 10 cm) in every coral reef around the world based on reef fish biomass observations. For all coral reef polygons, we predicted the biomass of reef fish (excluding pelagic or transient species) per unit area (kg/ha) under two alternative conditions: non-MPA and sustainable-use MPAs, while accounting for each site's own environmental and social covariates (Supplementary Fig. S2). Site covariates considered in this analysis included chlorophyll concentration, sea surface temperature mean, sea surface temperature range, nitrate concentration, wave exposure, reef area, shore distance, human population, market distance, human development index, government effectiveness, and fisheries management effectiveness (see Table S1 and Supplementary Material for detailed descriptions)^{52,57}. To account for variability in MPA effectiveness across countries (due to differences in management, staff capacity, state of the reef prior to MPA establishment, etc.), we also considered an interaction term between presence of sustainable-use MPAs and fisheries management effectiveness (FME) across nations²⁴. FME was calculated based on a survey with over 13,000 fisheries experts to assess the effectiveness of fisheries management regimes worldwide in 2008²⁴. In addition, we set ecoregions⁵⁸ as a random effect to account for the spatial structure of the data. Collinearity among covariates was examined based on bivariate correlations and variance inflation factors (all pairwise correlations were above 0.6 and VIF below 1.5), which led to the exclusion of selected environmental variables (pH, salinity, primary productivity, and minimum sea surface temperature) and social variables (land cover and fisher density), none of which were correlated or collinear with our variables of interest (sustainable-use MPAs and fisheries management effectiveness).”

And in the supplementary material:

“Fisheries management effectiveness: Effectiveness with which fisheries are being managed. Based on averaged scores on the scales of scientific robustness, policymaking

transparency, implementation capability, fishing capacity, subsidies, and access to foreign fishing. This study used a survey with over 13,000 fisheries experts to assess the effectiveness of fisheries management regimes worldwide in 2008 (Mora et al 2009).”

Reviewer #3:

Review_Nat Comms

Sustainable-use marine protected areas to improve human nutrition

This paper aims to show the nutritional impacts of increasing the global coverage of sustainable-use marine protected areas. It does this through a series of steps modeling increased production, increased catch, and increased nutritional intake across close to 2000 reef systems globally. This is an important issue for the policy, practice and the scientific communities to engage with. The manuscript was clearly written and methodologically strong. I have two big concerns with the manuscript as written and a few minor comments.

Big

The arc of this paper is based on the implicit assumption that MPAs (in this case SUMPAs) are the policy tool that needs to be used to improve the coastal nutrition inadequacy. While I am sympathetic to that assumption the manuscript needs to set it up as why that is the case AND also be honest about the other approaches. For example, if I were a more traditional fisheries management scientist/economist my first thought would be the this issue is really about Fisheries Management Effectiveness (FME). That is the problem to solve. The authors themselves show that where FME is high there less biomass boost from an MPA. First, it would be nice to see a statistical test around how these distributions differ. Second, how does one go right from the right to the idea that increased FME should not be evaluated as the nutrition intervention. That is an analysis I think that would really benefit fisheries management. E.g. “Improving FME gives us this nutritional boost, but when we add SUMPAs the boost is bigger... but only here, here and here.” By not acknowledging that there are these other avenues for nutrition improvements, the manuscript come off as having an agenda... that is eventually proven.

Thank you for the comment. In Figure 1, we show the effect of sustainable-use MPAs for locations with high (>0.5) estimated fisheries management effectiveness (FME) and low FME (<=0.5). Indeed, national-level fisheries management effectiveness has a strong effect on reef fish biomass, indicating that increasing FME has potentially large nutritional benefits. However, national-level top-down fisheries management strategies depend on strong management institutions. For locations with weaker management institutions, sustainable-use MPAs and other local-level management strategies can be more effective in providing fisheries benefits since it empowers local stakeholders and incentivizes resource

management, while considering local characteristics of the fishery and cultural traditions of coastal communities. In addition, we cannot expect high FME unless there is an actual sustainability and biodiversity conservation commitment (i.e. within many MPAs), specifically for complex fisheries in tropical coastal areas. To address this important comment:

- We Added a new paragraph in the introduction (lines 64-68):

“The integration of sustainable-use MPAs into the food and nutrition agendas of coral reef countries is often overlooked¹⁷. While good fisheries management holds the potential to enhance human nutrition⁶, its success is contingent upon the availability of high-quality data and robust management capacities. Sustainable-use MPAs emerge as a realistic and cost-effective intervention in various settings, offering a balance between conservation goals and sustainable resource utilization¹⁸. Moreover, these conservation measures play a crucial role in safeguarding the rights of local communities, protecting them from exploitation by non-local entities¹⁹. This not only underscores the significance of MPAs as a pivotal tool in sustainable resource management but also highlights their potential multifaceted impacts on both ecological conservation and human nutrition.”

- And added text related to the importance of FME in the discussion (lines 301-303):

“We consider sustainable-use MPAs as all protected areas where some form of fishing is permitted¹⁰. In addition to state-managed areas, this definition encompasses many types of area-based management strategies co-managed with or solely managed by local communities. Different from top-down fisheries management strategies, these and some other effective area-based conservation measures (OECMs) can empower local stakeholders and incentivize sustainable resource management, while considering local characteristics of the fishery and cultural traditions of coastal communities⁴². Several studies have shown that when communities are empowered and have secure rights to a fishery, there is greater incentive for successful fisheries management yielding nutritional benefits⁴³⁻⁴⁵. However, more research is needed to evaluate how different types of sustainable-use MPAs and restrictions affect biomass, catch, and human nutrition. Various other management measures (e.g., individual quotas) that are implemented outside of sustainable-use MPAs could also be effective at improving nutrient supply. Our analysis suggests that fisheries management effectiveness at a national level has a strong effect on reef fish biomass and can be important to providing broad nutritional benefits to coastal populations.”

This leads to my second concern if increasing FME can boost nutrition it does so at what cost? This is a very different paper than the ones the authors have written, but it begs the question about the cost of implementing SUMPAs. Why jump right to them? I think there are compelling reasons to do so but this is not addressed. Given the authors cite a really strong paper on the problems and inadequacies of MPA funding it would be good to elevate that part of the discussion. So why SUMPAs? What are the costs? Does the nutrition boost justify those costs cf other ways to increase nutrition? Again, there are many reasons an SUMPAs is a good idea and I think the authors should lay them out and then say... AND we are looking at the nutrition benefit.

This is a very important point. Nutritional benefits from sustainable-use MPAs should be viewed as an add-on benefit on top of the main goal of MPAs - which is marine conservation. We are not suggesting that MPAs should be implemented to solve nutritional problems since there could be other more cost-effective ways to do that. We are suggesting that MPAs can have important nutritional benefits on top of conservation, livelihoods, fisheries, etc. This is an important point that needs to be clear throughout the manuscript. Therefore, we added text throughout the manuscript to make this very clear, including:

- In the abstract (lines 35-36)

“Coral reef fisheries are a vital source of nutrients for thousands of nutritionally vulnerable coastal communities around the world. Marine protected areas are regions of the ocean designed to preserve or rehabilitate marine ecosystems and thereby increase reef fish biomass. Here, we evaluate the potential effects of expanding a subset of marine protected areas that allow some level of fishing within their borders (sustainable-use MPAs) to improve the nutrition of coastal communities. We estimate that, depending on site characteristics, expanding sustainable-use MPAs could increase catch by up to 20%, which could help prevent 0.3-2.85 million cases of inadequate micronutrient intake in coral reef nations. Our study highlights the potential add-on nutritional benefits of expanding sustainable-use MPAs in coral reef regions and pinpoints locations with the greatest potential to reduce inadequate micronutrient intake level. These findings provide critical knowledge given international momentum to cover 30% of the ocean with MPAs by 2030 and eradicate malnutrition in all its forms.”

- Introduction (Lines 73-74):

“Here, we quantified the potential add-on human nutritional benefits that can arise from sustainable-use MPA implementation through increases in local reef fish catch and consumption. We started by compiling information on coral reef fish populations and social and environmental conditions from 2,421 reef sites (1,117 no-take MPA, 804 sustainable-

use MPA, 500 non-MPA) in 53 countries to estimate the effect of these areas on standing reef fish biomass. We then used a Bayesian hierarchical model to estimate the expected standing reef fish biomass under non-MPA and sustainable-use MPAs conditions, accounting for other social (e.g., human population, market distance, fisheries governance, human development index) and environmental (e.g., productivity, depth, temperature, wave exposure) variables (see Supplementary Material for details). We then used the model to estimate: 1) the expected biomass and catch for all existing sustainable-use MPAs; 2) the potential biomass and catch associated with an expansion of sustainable-use MPAs to all non-MPA reefs; 3) the change in nutrient supply due to changes in coral reef fish catch; and 4) the potential changes in nutritional inadequacies associated with sustainable-use MPA expansion (see Supplementary Fig. M1 for details). We focused our analyses on zinc, iron, calcium, omega-3 long-chain polyunsaturated fatty acids docosahexaenoic acid (DHA) and eicosapentaenoic acid (EPA) (hereafter referred as DHA+EPA), and vitamins A and B₁₂, which are abundant in aquatic species and critical for human health⁶.”

- Discussion (lines 304-314):

“Nutritional benefits of sustainable-use MPAs should be viewed as an add-on benefit that supplements the marine conservation goals of MPAs. Compared to other measures to address malnutrition, sustainable-use MPAs can be costly and take a long time to provide desired benefits. Other measures such as nutritional supplements and agriculture development can be faster and more cost-effective to address malnutrition⁴⁶. However, sustainable-use MPAs represent a synergistic policy that can meet multiple objectives: conservation, food security, ecosystem resilience, property rights allocation, conflict alleviation, etc., while other targeted policies are usually for single purposes. In addition, sustainable-use MPAs can act as a safety net during food supply shocks, providing a reliable nutrient source easily accessible by local communities. Moreover, sustainable-use MPAs can be important to mitigate future nutritional loss due to anthropogenic actions such as overfishing and climate change⁸.”

Finally (but related), the argument of the paper is that poor coastal areas near reefs need to eat more fish, so they need to catch more fish, so they need SUMPAs. Aren't they trying to catch more fish? And haven't they been trying to? And is that one of the reasons reefs are depleted? (of course there are other drivers of overfishing ... industrial, invasives, dynamiting etc...). So the SUMPAs need to be placed in places already productive enough for locals but to ensure their longer term sustainability, OR in places that need some rebuilding. If the former then there is no nutrition boost (but there is the mitigation of nutrition loss in the future which is good). If the latter, then COST comes up again. The cost of rebuilding the ecosystem and stocks etc... These costs could be financial, social, but also the loss of off-take, temporary closures, altered off-take zones and timing. The manuscript

deals with this problem in one line (ln 260). This is a significant problem in marine conservation and should be more deeply addressed. This is an especially acute problem for those who “fishing rhymes with poverty” who cannot afford temporary loss of catch.

This is another great point that needs to be addressed throughout the manuscript. First, SUMPAs can be key to mitigate future nutrition loss. We added text to make this clear on lines 264-265, 313-314, 319-320. For example (lines 313-314):

“Nutritional benefits of sustainable-use MPAs should be viewed as an add-on benefit that supplements the marine conservation goals of MPAs. Compared to other measures to address malnutrition, sustainable-use MPAs can be costly and take a long time to provide desired benefits. Other measures such as nutritional supplements and agriculture development can be faster and more cost-effective to address malnutrition⁴⁶. However, sustainable-use MPAs represent a synergistic policy that can meet multiple objectives: conservation, food security, ecosystem resilience, property rights allocation, conflict alleviation, etc., while other targeted policies are usually for single purposes. In addition, sustainable-use MPAs can act as a safety net during food supply shocks, providing a reliable nutrient source easily accessible by local communities. Moreover, sustainable-use MPAs can be important to mitigate future nutritional loss due to anthropogenic actions such as overfishing and climate change⁸.”

Second, short-term economic, nutritional costs of rebuilding stocks is critical for successful implementation of SUMPAs and we agree that this needs to be discussed in greater depth. We added text to the discussion on this (see lines 286-288):

“Our model predictions are based on the current performance of sustainable-use MPAs. To restrict catch, these areas use an array of fisheries management tools, including gear restrictions, access rights, size limits, temporal closures, bag limits, and more⁹. Some sustainable-use MPAs may be zoned for multiple uses, potentially containing small no-take areas, which may benefit fished areas through spillover³⁸. Because our results depend on the actual performance of sustainable-use MPAs, if management of these areas improves (through more investment, management capacity, planning, community participation, etc.), potential biomass increases and consequential nutritional gains could be even greater. Understanding how close sustainable-use MPAs are to their maximum sustainable yield can provide a better estimate of their true potential to provide nutritional benefits. In some cases, reef stocks could be managed to maximize specific nutritional yields, tailored to the needs of local human populations³⁹. However, for fish stocks experiencing high fishing pressure and depletion below the biomass supporting maximum sustainable yields, recovery will likely require a short-term decrease in catch to obtain long-term gains^{40,41}. Therefore, it is important to consider the impacts of potential short-term economic and

nutritional costs to achieve predicted long-term nutritional benefits. Given that fish recovery may require restricting catch for an extended period, policy makers should seek to mitigate impacts of regulations on nutritional security in highly dependent populations."

So, in all, this is a good piece of work and will be an addition to the literature.

Minor Comments

Last sentence of abstract is confusing as written.

We appreciate this comment because this was confusingly written in the original submission. We edited the sentence by breaking it into two sentences:

"Our study highlights the potential add-on nutritional benefits of expanding sustainable-use MPAs in coral reef regions and pinpoints locations with the greatest potential to reduce inadequate micronutrient intake level. These findings provide critical knowledge given international momentum to cover 30% of the ocean with MPAs by 2030 and eradicate malnutrition in all its forms."

Ln 40 – A bit of a slight of hand to go from coastal fisheries to coral reefs. I think it would be better to slow down and explain in two more sentences this relationship/the overlaps/ this importance of the latter to the former.

Thank you for the comment. We have revised this text and added a sentence to make these connections clearer (lines 44-45):

"Over 2 billion people are unable to access safe, nutritious, and sufficient supplies of food, which threatens human health globally¹. Millions of coastal residents in tropical developing countries rely on fisheries resources as a vital source of minerals, vitamins, and fatty acids^{2,3}, and are particularly vulnerable to nutritional deficiencies⁴⁻⁶. For many of these coastal populations, coral reef systems are a critical source of nutrients and livelihoods⁷. Yet, coral reefs around the world are being severely degraded by pollution, overfishing, and climate change, imperiling marine biodiversity and human health⁸. Policies governing coral reefs that attempt to address these threats not only shape the future of these ecosystems but also the health of people who depend upon them."

Ln 61 – "increased production of seafood stocks" ? Is this the best way to characterize this?

This is a good point. We changed the sentence to refer specifically to increased catch of reef fish.

Contrast is Figure 1 A, B, C – not existent in my download.

We apologize for this difficulty. We uploaded a different image and hope it is more visible.

Fig 2. “error lines” ? what are they? Where are they?

Error lines represent the variability in the impact across reef polygons within each country. We changed the text in the caption to make this clearer (see figure 2 caption lines 134-135):

“Fig. 2. Estimated catch and nutritional benefits from existing sustainable-use MPAs. Estimated catch effects in existing sustainable-use MPAs relative to expected catch under non-MPA conditions, plotted on a log-scale for country populations within 10km of sustainable-use MPAs. Error lines represent the variability across reef polygons within each country. Inadequate nutrient intake is the average prevalence across key nutrients found in aquatic species in 2017 (iron, EPA+DHA, calcium, zinc, and vitamins A and B₁₂). Points in gray represent countries with no data on prevalence of nutrient intake. Three-letter abbreviations represent selected alpha-3 ISO country codes.”

Lines 277 on... - Climate Change platitudo comes in out of nowhere, is superficial and its import and interaction with the MPA discussion – poorly supported.

Thank you for the comment. We agree that the flow here can be improved. We added another paragraph before this discussion to improve flow. See lines 313-314:

“Nutritional benefits of sustainable-use MPAs should be viewed as an add-on benefit that supplements the marine conservation goals of MPAs. Compared to other measures to address malnutrition, sustainable-use MPAs can be costly and take a long time to provide desired benefits. Other measures such as nutritional supplements and agriculture development can be faster and more cost-effective to address malnutrition⁴⁶. However, sustainable-use MPAs represent a synergistic policy that can meet multiple objectives: conservation, food security, ecosystem resilience, property rights allocation, conflict alleviation, etc., while other targeted policies are usually for single purposes. In addition, sustainable-use MPAs can act as a safety net during food supply shocks, providing a reliable nutrient source easily accessible by local communities. Moreover, sustainable-use MPAs can be important to mitigate future nutritional loss due to anthropogenic actions such as overfishing and climate change⁸.

Climate change impacts on reef ecosystems create an uncertain future for millions of people who depend on reef fisheries for nutrition and livelihood⁸. Without actions (such as sustainable-use MPAs) to ensure sustainability of catch into the future, there could be

significant losses of nutritional benefits in the coming years, with important implications for public health. Therefore, it is not only the prevention of inadequate intake and the supporting benefits of sustainable-use MPAs that can be important, but also the buffering against the risk of future loss in climate-vulnerable systems. Without conservation action in the present, the risk of future negative nutritional impacts is inevitable in the long-term⁸.”

Reviewers' Comments:

Reviewer #1:

Remarks to the Author:

I have now read the revised version of the manuscript as well as the other reviewers' comments and authors' responses. The authors' have addressed all of my comments thoughtfully and sufficiently. It seems they have done the same with the other reviewers' comments. Overall, the additional text regarding methods and estimation assumptions will be useful for the reader. Furthermore, the findings of the study have been slightly tempered with a bit more contextualization within reality, for example the magnitude of the potential nutritional benefit relative to the cost of MPAs as a management intervention (re-emphasizing nutrition as an add-on benefit I think is important) and further emphasizing the potential need to decrease catch in the short term for long-term sustainability. I think this latter point, which was brought up by another reviewer, is critical, and I wish there could have been some analysis of the extent to which we might expect these short-term falls in nutrition as a necessary path to sustainable nutrient provision from reef fisheries, as well as the timeframe, because otherwise readers may have the lingering question: would conversion to multi-use MPAs actually be a nutritional liability in the short term (and how long is the short term)? However, I realize this was beyond the scope, and the existing findings showing increasing catch from these MPAs utilized in this analysis are reassuring from this perspective. I think that the abstract also accurately represents the findings. I do wonder if the title should be revised to reflect the relatively small magnitude (but not necessarily small importance) of nutritional gains and characterization of these gains as add-on benefits throughout the text.

Reviewer #2:

Remarks to the Author:

With the changes in response to my comments and those of the other reviewers, the manuscript is now much clearer in explaining the methodology and outlining the key results. I especially like the updated version of Fig. 4. I am satisfied with the updated manuscript and recommend it for publication in Nature Communications.

Response to Reviewers

REVIEWERS' COMMENTS

Reviewer #1 (Remarks to the Author):

I have now read the revised version of the manuscript as well as the other reviewers' comments and authors' responses. The authors' have addressed all of my comments thoughtfully and sufficiently. It seems they have done the same with the other reviewers' comments. Overall, the additional text regarding methods and estimation assumptions will be useful for the reader. Furthermore, the findings of the study have been slightly tempered with a bit more contextualization within reality, for example the magnitude of the potential nutritional benefit relative to the cost of MPAs as a management intervention (re-emphasizing nutrition as an add-on benefit I think is important) and further emphasizing the potential need to decrease catch in the short term for long-term sustainability. I think this latter point, which was brought up by another reviewer, is critical, and I wish there could have been some analysis of the extent to which we might expect these short-term falls in nutrition as a necessary path to sustainable nutrient provision from reef fisheries, as well as the timeframe, because otherwise readers may have the lingering question: would conversion to multi-use MPAs actually be a nutritional liability in the short term (and how long is the short term)? However, I realize this was beyond the scope, and the existing findings showing increasing catch from these MPAs utilized in this analysis are reassuring from this perspective. I think that the abstract also accurately represents the findings. I do wonder if the title should be revised to reflect the relatively small magnitude (but not necessarily small importance) of nutritional gains and characterization of these gains as add-on benefits throughout the text.

We thank the reviewer for these comments and appreciate all the feedback that greatly improved the manuscript. We agree with the reviewer that an analysis regarding timeframe would be a great addition to the analysis. However, as pointed out by the reviewer, this was beyond the scope of this project. We are grateful for the suggestion to review the title, but after a debate among all co-authors, we have decided to keep the original title.

Reviewer #2 (Remarks to the Author):

With the changes in response to my comments and those of the other reviewers, the manuscript is now much clearer in explaining the methodology and outlining the key results. I especially like the updated version of Fig. 4. I am satisfied with the updated manuscript and recommend it for publication in Nature Communications.

We thank the reviewer for all valuable comments and thoughtful feedback. We agree that Fig. 4 was greatly improved in the updated manuscript, and we thank the reviewer for the suggestion.